# Long-Text-to-Image Generation via Compositional Prompt Decomposition

**Jen-Yuan Huang**[1†]    **Tong Lin**[1*†]    **Yilun Du**[2*]
[1]Peking University    [2]Harvard University

## Abstract

While modern text-to-image (T2I) models excel at generating images from intricate prompts, they struggle to capture the key details when the inputs are descriptive paragraphs. This limitation stems from the prevalence of concise captions that shape their training distributions. Existing methods attempt to bridge this gap by either fine-tuning T2I models on long prompts, which generalizes poorly to longer lengths; or by projecting the oversize inputs into normal-prompt space and compromising fidelity. We propose **P**rompt **R**efraction for **I**ntricate **S**cene **M**odeling (*PRISM*), a compositional approach that enables pre-trained T2I models to process long sequence inputs. PRISM uses a lightweight module to extract constituent representations from the long prompts. The T2I model makes independent noise predictions for each component, and their outputs are merged into a single denoising step using energy-based conjunction. We evaluate PRISM across a wide range of model architectures, showing comparable performances to models fine-tuned on the same training data. Furthermore, PRISM demonstrates superior generalization, outperforming baseline models by **7.4%** on prompts over 500 tokens in a challenging public benchmark. Project website: `jy-joy.github.io/PRISM`.

## 1 Introduction

Compositionality is fundamental to human intelligence—the ability to understand novel concepts by decomposing them into familiar primitives and to build complex systems from simple components. This "divide and conquer" strategy is also common in creative activities. An artist, for instance, rarely materializes an intricate scene holistically. Instead, they might independently perfect the rendering of a rustic wooden house and the surrounding trees, ensuring each element is realized with care before integrating them into a cohesive whole. In stark contrast, text-to-image (T2I) models (Rombach et al., 2022) attempt to render the entire scene simultaneously in a single, monolithic process.

This paradigm works well for concise prompts but falters when the input becomes a descriptive paragraph. While a model may excel at rendering "a house in the middle of a forest", it often fails when the prompt expands, detailing the terracotta roof tiles, the weathered white panels of the house, and the striking contrast cast by the afternoon sun. This failure stems from a fundamental conflict between the nature of long-form text and the models' training paradigm. T2I models are predominantly trained on vast datasets of images paired with short, concise captions. They learn to map phrases to visual features but are undertrained on interpreting the narrative flow and distributed details of a paragraph (Bai et al., 2024). Even modern models using powerful text encoders struggle on these inputs, missing more than half of the specified objects (Jiao et al., 2025).

Existing methods attempt to bridge this gap through two strategies (Figure 2.a & b). The most direct approach fine-tunes the T2I model on long-captioned data (Bai et al., 2024; Wu et al., 2025b). While being effective within the training lengths, extrapolating to even longer prompts remains challenging, and the tuned models risk "catastrophic forgetting" of their pre-trained knowledge. The second strategy adopts projection-based methods to map the long prompt into the effective context window of the pre-trained models (Hu et al., 2024; Liu et al., 2025). The information bottleneck sacrifices the very details that make the long prompts compelling. These limitations reveal an open question: *how can we utilize a model's existing knowledge of short prompts to render long, intricate paragraphs?*

---

[*]Corresponding to `lintong@pku.edu.cn` and `ydu@seas.harvard.edu`.
[†]State Key Laboratory of General Artificial Intelligence, School of Intelligence Science and Technology.

| Baseline | Component 1 | Component 2 | Component 3 | Component 4 | Composition |
|---|---|---|---|---|---|

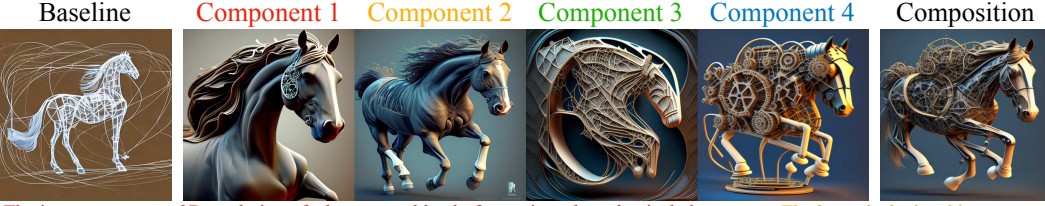

The image presents a 3D rendering of a horse … a blend of organic and mechanical elements … The horse is depicted in a state of motion, with its mane and tail flowing behind it … The horse's body is composed of a network of lines and curves … This intricate design is further emphasized by the presence of gears and other mechanical components …

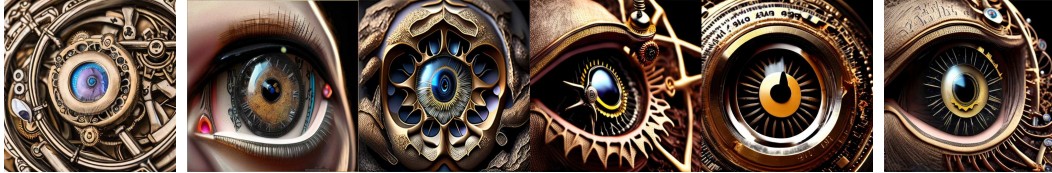

A hyper-detailed, macro shot of a human eye … thread-like veins in the sclera are reimagined as fine, coiling copper wires … presented not as an organ of sight, but as a gateway to a lost world … The iris is a masterfully crafted, antique horological mechanism … interlocking gears and cogs made from polished brass, copper, and tarnished silver …

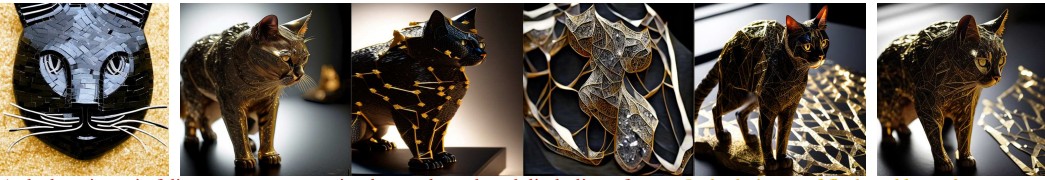

A sleek, enigmatic feline … rests upon a simple, unadorned, and dimly lit surface … Its body is not of flesh and bone, but meticulously sculpted from a complex lattice of polished, interlocking obsidian shards … defined by the sharp, clean edges of these volcanic glass fragments, giving its natural curves a subtle, geometric undertone … Glimmering veins of molten gold … trace the contours of the cat's muscles and skeleton, outlining its elegant spine …

Figure 1: **Compositional Long Prompt Decomposing.** We decompose the long prompt into semantic components, each depicting parts of the input content. At each denoising step, model outputs for each components are composed into a single noise prediction, rendering the entire paragraph as a whole.

Drawing inspiration from the compositional generative modeling to generalize individual models to new tasks beyond their initial capacity (Du & Kaelbling, 2024; Du et al., 2023), we advocate the idea of compositionality for long-text-to-image generation. Rather than forcing the model to process an out-of-distribution long sequence, we decompose it into a set of manageable components (Figure 2.c). The final image is generated from a factorized distribution, where constituents are simpler factors conditioned on their respective components. The compositional strategy offers two unique advantages: first, it allows the pre-trained model to operate within its domain of expertise, thereby eliminating the need for fine-tuning and preserving the pre-trained knowledge. On the flip side, it ensures higher fidelity by distributing information across multiple, dedicated components.

The central challenge lies in robust decomposition; a simple linguistic split loses global context in each component, resulting in inconsistent scene generation. We propose **P**rompt **R**efraction for **I**ntricate **S**cene **M**odeling (*PRISM*), a universal framework that learns this decomposition directly in the conditional input space. PRISM adopts a lightweight module to decompose the long-prompt encoding into constituent representations. The T2I model makes independent noise predictions on each of the extracted representations, which are merged into a single output using energy-based conjunction. The decomposition module is optimized in an unsupervised manner guided by the frozen T2I model that learns to extract components that are not only interpretable by the pre-trained model, but also yield coherent composite scene rendering as demonstrated in Figure 1. We evaluate PRISM across a wide range of T2I architectures, demonstrating comparable performance to models fine-tuned on the same training dataset. Crucially, PRISM achieves superior generalization as prompt length increases, outperforming the baselines by an average of **7.4%** on prompts over 500 tokens. Our contributions can be summarized as:

1. We propose a compositional strategy enabling pre-trained T2I models to process out-of-distribution descriptive paragraphs effectively.

2. We introduce PRISM, a universal framework utilizing a lightweight decomposition module learning to decompose long-prompt encodings without explicit decomposition supervision.

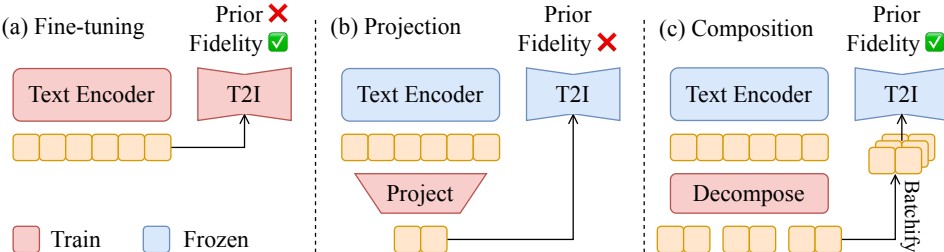

Figure 2: **Long-Text-to-Image Generation Strategies.** (a) Fine-tuning the pre-trained T2I model on long prompts; (b) Projecting long prompts to the compact semantic window; (c) Instead of forcing alignment, we decompose long prompts into components for compositional generation.

3. We conduct extensive experiments across various pre-trained T2I models, demonstrating PRISM's superior generalization to prompts longer than those seen during training.

## 2 RELATED WORK

### 2.1 LONG-TEXT-TO-IMAGE GENERATION

Diffusion models (Ho et al., 2020; Sohl-Dickstein et al., 2015) have significantly propelled visual generation. Integrated with text conditioning, these models can generate images with unprecedented diversity and quality from natural language descriptions (Rombach et al., 2022; Ramesh et al., 2022). Recent progress in model architecture (Peebles & Xie, 2023) and theoretical foundations (Liu et al., 2022b; Lipman et al., 2022) have enabled T2I models to scale to billions of parameters (Esser et al., 2024; Batifol et al., 2025). Despite this progress, a key limitation remains their difficulty in interpreting long, descriptive paragraphs (Jiao et al., 2025). This challenge often stems from the fixed context window of the text encoders (e.g., CLIP (Radford et al., 2021)), which can be overcome by using more powerful language models (LMs) (Zhao et al., 2024; Liu et al., 2025). However, adapting to the new input takes intensive tuning. An efficient strategy involves projecting the LM representations into the T2I model's original text embedding space (Hu et al., 2024; Liu et al., 2025). To systematically evaluate performance on this task, DetailMaster (Jiao et al., 2025) introduces a rigorous benchmark consists of intricate prompts with an average length of 284.9 tokens depicting complex scenes with multiple objects. It also provides a comprehensive, multi-stage evaluation pipeline leveraging multimodal models to analyze visual details.

### 2.2 COMPOSITIONAL GENERATIVE MODELING

Our work builds on the principle of compositional generative modeling, which constructs complex generative systems by combining simpler, specialized models rather than training a single monolithic one (Du & Kaelbling, 2024; Garipov et al., 2023). Conceptually, this approach treats each model as a soft constraint and uses optimization techniques to find outputs that have a high likelihood across all constituent models (Du et al., 2023; Yang et al., 2023). A key advantage of this approach is its data efficiency and generalization capability; by learning simpler, factorized distributions, a compositional system can generate valid samples for combinations of patterns unseen during training (Mahajan et al., 2024). In vision domain, compositional methods enable the generation of novel images with blended features (Du et al., 2020). For instance, composing T2I diffusion model outputs on different text prompts leads to a sample that is collectively described by all prompts (Liu et al., 2022a; Bradley et al., 2025; Bar-Tal et al., 2023; Yang et al., 2024). It is also possible to train a compositional generative system as a whole. This allows each constituent model to learn a compositional factor from data, which can then be recombined to synthesize novel combinations (Su et al., 2024; Liu et al., 2023). Similarly, we approach the challenge of long-text-to-image generation through a compositional lens, aiming to identify and model the compositional factors within a complex text prompt.

## 3 METHODOLOGY

Our approach achieves long-text-to-image generation by reframing it as a compositional task. Instead of training a monolithic model to interpret an entire paragraph, we decompose the paragraph into

a set of components that a pre-trained T2I model can readily understand. The final image is then synthesized by composing the model's outputs for each component, a technique made possible by the insight that diffusion models can be treated as composable energy-based models.

### 3.1 PRELIMINARIES: COMPOSING DIFFUSION MODELS

**Text-to-Image Diffusion Generation.** A T2I diffusion model, $\epsilon_\theta(\boldsymbol{x}_t, t, \boldsymbol{c})$, generates an image $\boldsymbol{x}$ conditioned on a text prompt $\boldsymbol{c}$ by progressively denoising the input to decreased noise levels $\{\sigma_t\}_{t=1}^T$ (Ho et al., 2020). The model is trained to predict the noise $\epsilon_t$ added to an image $\boldsymbol{x}$ at timestep $t$. Generation begins with pure Gaussian noise, $\boldsymbol{x}_T \sim \mathcal{N}(\boldsymbol{0}, \sigma_T^2 \boldsymbol{I})$, which the model iteratively refines by subtracting the predicted noise at each step. This process corresponds to score-based modeling (Song et al., 2020b), where the predicted noise is proportional to the time-dependent score function (the gradient of the log likelihood): $\epsilon_\theta \propto -\nabla_{\boldsymbol{x}_t} \log p_t(\boldsymbol{x}_t | \boldsymbol{c})$. Generation can thus be viewed as a form of Langevin dynamics (Du & Mordatch, 2019),

$$\boldsymbol{x}_{t-1} = \boldsymbol{x}_t + \frac{\sigma_t^2}{2} \nabla_{\boldsymbol{x}_t} \log p_t(\boldsymbol{x}_t) + \sqrt{\sigma_t}\epsilon. \tag{1}$$

where the learned score function at each timestep gradually guides a sample toward a high-density region of the target data distribution $p(\boldsymbol{x}|\boldsymbol{c})$.

**Energy-Based Compositionality.** The score-based view of diffusion models reveals a connection to Energy-Based Models (EBMs). An EBM defines a probability density via an unnormalized energy function, $p_\theta(\boldsymbol{x}) \propto e^{-E_\theta(\boldsymbol{x})}$, and uses the gradient of this energy function with Langevin dynamics for generation. A key advantage of EBMs is their inherent compositionality; sampling from a product of distributions is as simple as summing their energy functions (Du et al., 2020; Du, 2025):

$$p_{compose}(\boldsymbol{x}) \propto \prod_i p_\theta^i(\boldsymbol{x}) \propto e^{-\sum_i E_\theta^i(\boldsymbol{x})}, \tag{2}$$

yielding sample with high-likelihood across all constituent EBMs. As demonstrated by prior work, this logic can be extended to diffusion generation by drawing an line between the diffusion model and the gradient of an implicit energy function, $\epsilon_\theta \approx \nabla_{\boldsymbol{x}_t} E_\theta(\boldsymbol{x}_t)$. To sample from the product of two distributions conditioned on $\boldsymbol{c}_1$ and $\boldsymbol{c}_2$, one can simply sum their respective noise predictions,

$$\boldsymbol{\epsilon}_{composed}(\boldsymbol{x}_t, t) = \boldsymbol{\epsilon}_\theta(\boldsymbol{x}_t, t, \boldsymbol{c}_1) + \boldsymbol{\epsilon}_\theta(\boldsymbol{x}_t, t, \boldsymbol{c}_2) \propto \nabla_{\boldsymbol{x}_t} \log\left(p_t(\boldsymbol{x}_t|\boldsymbol{c}_1) \cdot p_t(\boldsymbol{x}_t|\boldsymbol{c}_2)\right), \tag{3}$$

in the score function of Equation 1. This operation, known as **concept conjunction** (Liu et al., 2022a), forms a new composite score that guides the generation process toward an image satisfying both prompts simultaneously. Notably, the synthesized sample won't have to be presented in either of the training data in $p(\boldsymbol{x}|\boldsymbol{c}_1)$ and $p(\boldsymbol{x}|\boldsymbol{c}_2)$. This principle allows us to construct novel scenes from familiar concepts, laying the cornerstone for our approach.

### 3.2 COMPOSITIONAL LONG-TEXT-TO-IMAGE GENERATION

The domain gap in input prompts is the core challenge of long-text-to-image generation. A descriptive paragraph, $\boldsymbol{C}$, is fundamentally an out-of-distribution input for pre-trained T2I model $\epsilon_\theta(\boldsymbol{x}_t, t, \boldsymbol{c})$. These models are trained on vast datasets like LAION (Schuhmann et al., 2022), which is dominated by short, label-like captions $\boldsymbol{c}$. Therefore the models primarily learn to map keywords and short-phrases to visual features, lack the ability of narrative comprehension. Our central hypothesis is that the complex conditional distribution $p(\boldsymbol{x}|\boldsymbol{C})$ described by the paragraph $\boldsymbol{C}$ can be effectively approximated by factorizing into a set of simpler distributions: $p(\boldsymbol{x}|\boldsymbol{C}) \propto \prod_i^N p(\boldsymbol{x}|\boldsymbol{c}_i)$, where each constituent distribution $p(\boldsymbol{x}|\boldsymbol{c}_i)$ is conditioned on a component $\boldsymbol{c}_i$. Intuitively, a paragraph can be abstracted as a collection of phrases with each capturing a distinct feature. Leveraging the concept conjunction principle in Equation 3, we can construct a long-text-to-image generation model by composing a same pre-trained T2I model $\epsilon_\theta$ with different components:

$$\boldsymbol{\epsilon_\theta}(\boldsymbol{x}_t, t, \boldsymbol{C}) = \sum_{i=1}^N \boldsymbol{\epsilon_\theta}(\boldsymbol{x}_t, t, \boldsymbol{c}_i). \tag{4}$$

This composite score leads to an image that is collectively described by the components $\{\boldsymbol{c}_1, \ldots, \boldsymbol{c}_N\}$. Because the decomposed components remain semantically concise, they can be readily processed by

Figure 3: **Compositional Long-Text-to-Image Generation Model.** PRISM decomposes the long-prompt encoding into constituent representations using a learnable decomposition module. At each denoising step, current noisy latent is first cloned by the number of decomposed components into a batch. The T2I model makes independent denoise predictions conditioned on each of the constituent textual representations. Finally, these denoise predictions are merged into a composite output through energy-based conjunction.

the pre-trained T2I model, avoiding resource-intensive fine-tuning. Furthermore, unlike projection-based methods that suffer from information loss, our factorized approach maintains high fidelity to the original paragraph by distributing its information across multiple components $\{c_1, \ldots, c_N\}$.

### 3.3 UNSUPERVISED LONG-PROMPT DECOMPOSITION

To obtain the decomposed components $\{c_1, \ldots, c_N\}$, one appealing option is to utilize LLMs to analyze and break down the paragraph. However, Equation 3 lacks explicit spatial control over each component, resulting in global context inconsistency and local concept blending. We propose to learn such decomposition directly in the textual representation space via a trainable decomposition ($\psi$) module. The T2I model utilizes the module's output to form the noise prediction as per Equation 4. Crucially, the entire composed model is trained end-to-end in an unsupervised manner, using the diffusion loss calculated on the composite score,

$$\mathcal{L}(\psi) = \mathbb{E}_{\boldsymbol{x},t}\left[\left\|\sum_{i=1}^{N} \boldsymbol{\epsilon_\theta}(\boldsymbol{x}_t, t, \boldsymbol{c}_i) - \boldsymbol{\epsilon}\right\|^2\right], \boldsymbol{\psi}(\boldsymbol{C}_{LM}) = \{\boldsymbol{c}_1, \ldots, \boldsymbol{c}_N\}. \tag{5}$$

By training on the frozen T2I model, PRISM learns to distribute the information into components $\{c_i\}$ that are explicitly optimized for the compositional generation process in Equation 3. This end-to-end training allows the learned decomposition to effectively capture spatial relationships and global attributes (e.g., lighting, style) that are critical for consistency but lost in linguistic splitting.

### 3.4 IMPLEMENTATION DETAILS

Our compositional approach enables T2I models that are predominantly trained on concise captions to process long sequence inputs outside their training data distributions. Notably, PRISM is a general framework not constrained to any specific model family or architecture. Based on the text encoders employed in different T2I models, here we present two PRISM module designs that we find efficient for extracting the constituent representations from long-prompt text encodings.

**Bidirectional Text Encoder.** For text encoders with a bidirectional attention mechanism (Vaswani et al., 2017), like the T5 (Raffel et al., 2020) encoder employed by Stable Diffusion-3.5 (Peebles & Xie, 2023) and FLUX (Batifol et al., 2025), we implement the decomposition module as a Querying Transformer (Li et al., 2023). Specifically, we adopt $N$ learnable vectors, each of size $L \times D$ where $L$ is the vector length and $D$ is the hidden dimension. These vectors are passed to the decomposition module as queries with the long-prompt encoding $C_{LM}$ as the keys and values (see Figure 10). Guided by the compositional diffusion loss in Equation 5, these queries thus learn to extract distinct semantic components that are optimal for compositional generation.

**Causal Language Model.** For T2I models built on causal LLM text encoders, such as Qwen-Image (Wu et al., 2025a) with Qwen2.5-VL (Bai et al., 2025), we leverage the LLM's powerful reasoning capability to output constituent representations directly in a sequence. As illustrated in Figure 11, we first replicate the input tokens by $N$ times, concatenating them with a special trainable token $\langle|\texttt{comp}_i|\rangle$ prepend to the $i, i \in N$ prompt tokens segment into a single, expanded sequence as the text encoder input. Instead of using a separate decomposition module, we apply Low Rank Adapter (LoRA) (Hu et al., 2022) to the text encoder, training them to reason over the expanded sequence and extract distinct semantic components for compositional generation guided by the compositional diffusion loss in Equation 5.

Table 1: **Evaluation Results on the DetailMaster Benchmark** (Jiao et al., 2025)**.** Quantitative comparisons are conducted within two groups: Long-Text-to-Image generation methods built on StableDiffusion-1.5, and state-of-the-art baselines including SDXL, StableDiffusion-3.5, FLUX and Qwen-Image. Numbers are reported in percentage accuracies and the best results in each group are marked in **Bold**.

| Model | Character Presence | Character Attributes | | | Character Location | Scene Attributes | | | Spatial Relation |
|---|---|---|---|---|---|---|---|---|---|
| | | Object | Animal | Person | | Background | Light | Style | |
| *Long-Text-to-Image Methods Built on SD-1.5* | | | | | | | | | |
| StableDiffusion-1.5 | 19.12 | 84.40 | 76.62 | 80.73 | 8.66 | 24.53 | 69.27 | 84.47 | 7.18 |
| LLM4GEN | 19.43 | 82.99 | 78.00 | 81.67 | 9.48 | 28.32 | 68.08 | 50.28 | 8.04 |
| LLM Blueprint | 18.69 | 81.40 | 76.25 | 76.53 | **18.40** | 56.69 | 83.28 | 67.07 | 14.16 |
| ELLA | 25.57 | 82.38 | 78.75 | 80.33 | 15.04 | 69.15 | 83.12 | 44.17 | 15.17 |
| LongAlign | 25.88 | 85.54 | 83.28 | 83.85 | 14.12 | 78.60 | 87.33 | 70.49 | 21.24 |
| ***PRISM*** | **28.21** | 84.78 | 83.24 | 84.54 | 16.57 | 82.45 | **92.48** | 64.10 | 20.88 |
| ***PRISM w/ tuning*** | 25.99 | **86.05** | **86.21** | **86.16** | 16.21 | **90.96** | 91.16 | **84.93** | **24.47** |
| *State-of-the-Art T2I Models with Modern Architectures* | | | | | | | | | |
| ParaDiffusion(*SDXL*) | 28.63 | 87.40 | 85.34 | 84.66 | 20.62 | 84.83 | 93.59 | 72.16 | 25.95 |
| StableDiffusion-3.5 | 39.01 | 87.60 | 87.57 | 89.55 | 31.91 | 93.82 | 92.53 | 95.31 | 39.36 |
| FLUX-Dev. | 42.02 | 91.14 | 89.61 | 90.23 | 38.18 | **95.73** | 96.91 | 95.28 | 44.94 |
| Qwen-Image | 40.46 | 90.21 | 89.13 | 91.29 | 40.14 | 92.00 | 96.93 | 91.53 | 47.02 |
| ***PRISM-Qwen*** | **46.84** | **91.55** | **90.36** | **93.53** | **41.49** | 94.62 | **97.32** | **95.62** | **49.23** |

Table 2: **Quantitative Comparisons of Image Generation Quality.** We employ preference models for assessing images in terms of prompt alignment and human aesthetics. Best results are marked in **Bold**.

| Model | CLIPScore | DenScore | PickScore | VQAScore | HPSv3 |
|---|---|---|---|---|---|
| *Long-Text-to-Image Methods* | | | | | |
| ELLA | 30.89 | 20.34 | 20.72 | 73.30 | 6.78 |
| LongAlign | **33.43** | **22.35** | 24.43 | 82.01 | **13.26** |
| ***PRISM-SD1.5*** | 32.56 | 22.24 | **24.50** | **83.22** | 13.03 |
| *State-of-the-Art T2I Models* | | | | | |
| StableDiffusion-3.5 | **34.97** | 22.37 | 21.63 | 86.12 | **13.39** |
| FLUX-Dev. | 33.30 | 22.56 | 21.89 | 86.19 | 13.17 |
| Qwen-Image | 33.85 | 22.25 | 20.98 | 85.02 | 8.56 |
| ***PRISM-Qwen*** | 34.12 | **22.93** | **22.04** | **86.21** | 12.05 |

## 4 EXPERIMENT

### 4.1 EXPERIMENTAL SETUP

**Training.** We implement our compositional approach on two pre-trained T2I models to demonstrate generalizability across varying architectures. *PRISM-SD1.5* is built on the widely-used Stable Diffusion-1.5 (SD-1.5) (Rombach et al., 2022) backbone. Consistent with prior work (Hu et al., 2024), we replace the original CLIP text encoder with T5-XL (Raffel et al., 2020) to accommodate the token length of descriptive paragraphs. We also develop *PRISM-Qwen* on Qwen-Image (Wu et al., 2025a) to validate our approach on a large-scale modern architecture.

We adopt the training dataset provided in LongAlign (Bai et al., 2024), comprising approximately 2 million images re-captioned by LLaVA-Next (Liu et al., 2024) or ShareCaptioner (Chen et al., 2024). We adopt an AdamW optimizer (Loshchilov & Hutter, 2017) with batch size 192 and learning rate $1.0e^{-5}$. This training takes about 20 hours on 4 A100 GPUs. PRISM-Qwen applies LoRA on the text encoder and is trained for 2,500 steps with batch size 24 and learning rate $5.0e^{-5}$.

**Evaluation.** We adopt the DetailMaster benchmark (Jiao et al., 2025) to comprehensively assess long-text-to-image performance. DetailMaster is a challenging benchmark consists of prompts with 284.89 tokens on average, evaluating generation quality across five dimensions. **Character Presence** verifies successfully generated characters, and **Character Attributes** measures whether their features match with the prompt. Accuracies are computed separately for object, animal, and person categories. **Character Locations** checks if they are positioned correctly. **Scene Attributes** evaluates adherence to overall scenic instructions of background, lighting, and style. Finally, **Spatial Relation** quantifies the model's ability to reflect the specified relationships between the characters.

StableDiffusion    LLM4GEN         ELLA         LongAlign      *PRISM*    *PRISM w/ tuning*

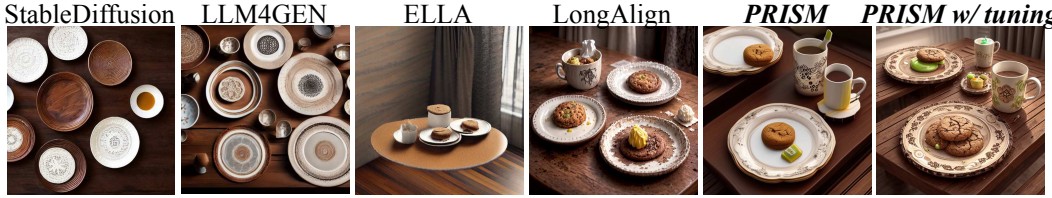

… In the center and on the left are two round, wavy side plates with black scratches on the sides and a doily pattern engraved on the plates … The cup, made of ceramic material, has a cylindrical shape with a handle and a textured surface …On the top right is a gray curtain, and on the upper left is a view of the lower part of a white wooden wall … The thick brown cookies crosscut at the top are positioned on top of the two round, wavy side plates, with one cookie on each plate … a candy with a yellow wrapper …

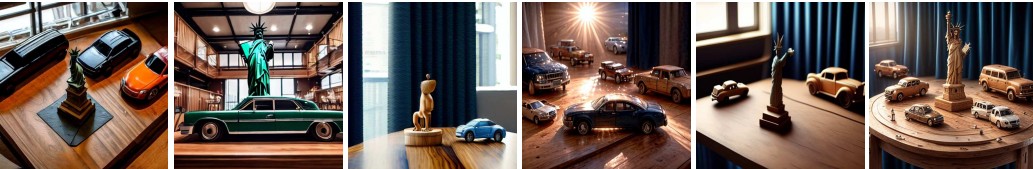

… An indoor top-down view of a wooden Statue of Liberty, which is positioned centrally on the table, covering a black marking on the table, on a wooden table with 3 wooden cars and 1 wooden limo next to it. … The wooden limo is placed to the left of the wooden Statue of Liberty, and the three wooden cars are arranged to the right … Behind the table is a dark blue curtain, through which sunlight is coming and shining down on the right side of the table, casting a soft glow and creating gentle shadows …

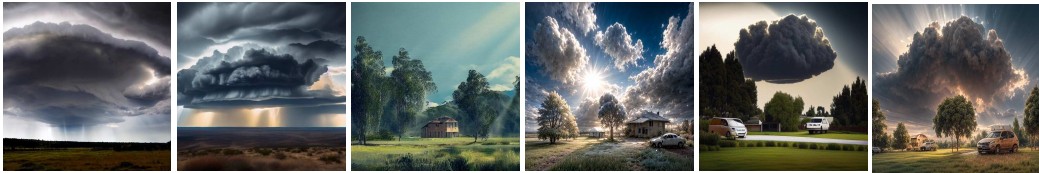

… A long-shot view of a slightly dark sky with a cumulonimbus forming in the clouds, allowing rays of sunlight to pierce through … the cumulonimbus cloud formation is a dark blue and gray, with soft, diffused sunlight breaking through the clouds … with the sun low on the horizon … A small house is visible in the distance; it has tan panels, and it has a white metal roof … Parked in the lower right part of the image in front of the house is a white sedan, situated between the house and the viewer …

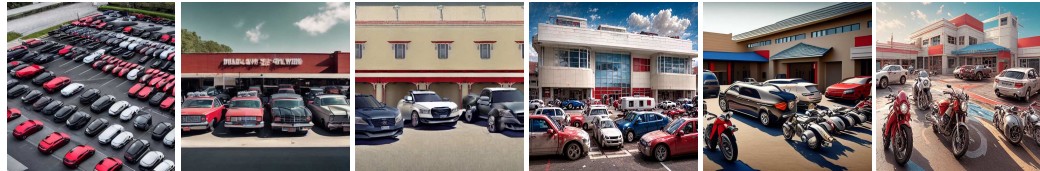

… A front view of a parking lot with several vehicles parked including two dark colored sedans in the middle part of the image and what appears to be six different motor bikes in front of them … a red motorbike with color red … a white motor bike in the left part of the image, a silver motor bike, another white motor bike, another silver motorbike and another silver motorbike … The parking has visible but faded white parking lines, and behind all of the vehicles are two handicap parking signs … background features a large cream-colored building with a blue roof and a red strip, partially obscured by the parked vehicles …

Figure 4: **Qualitative Comparisons with other Long-Text-to-Image Methods.** Images are generated from prompts in the DetailMaster benchmark, using different models built on StableDiffusion-1.5. Our PRISM accurately captures the intricate attributes and spatial relationships of various objects specified in the paragraphs. PRISM is also compatible with other model tuning methods to further enhance generation quality.

## 4.2 LONG-TEXT-TO-IMAGE GENERATION

**Long-prompt Following.** Table 1 summarizes the benchmark evaluations of DetailMaster, where we examine PRISM against specialized Long-Text-to-Image generation methods and SOTA baselines. PRISM-SD1.5 outperforms other methods by **2.33% on Character Presence** and **1.53% on Character Location**, demonstrating the PRISM's efficiency in processing descriptive paragraphs. Moreover, since the decomposed components remain in the pre-trained model's expected input space, PRISM can be used in conjunction with other fine-tuning methods to further enhance the results. Our model outperforms LongAlign across all metrics by 4.65% on average with the same tuning method.

**Enhancing SOTA Models.** Despite employing powerful text encoders (eg., T5-XXL or Qwen2.5-VL) in modern architectures, Table 1 also shows that more than a half of the characters described in the prompt are completely omitted by even the strongest FLUX model. Our compositional method can further enhance the performance of these SOTA models. For Qwen-Image that leverages an MLLM as its text encoder, our method improves the **Character Presence by 6.38%** and **Character Attributes by 1.60%** on average. This highlights the long-text-to-image generation as a fundamental challenge originates from the scarcity of long-captioned training data, instead of the text encoder.

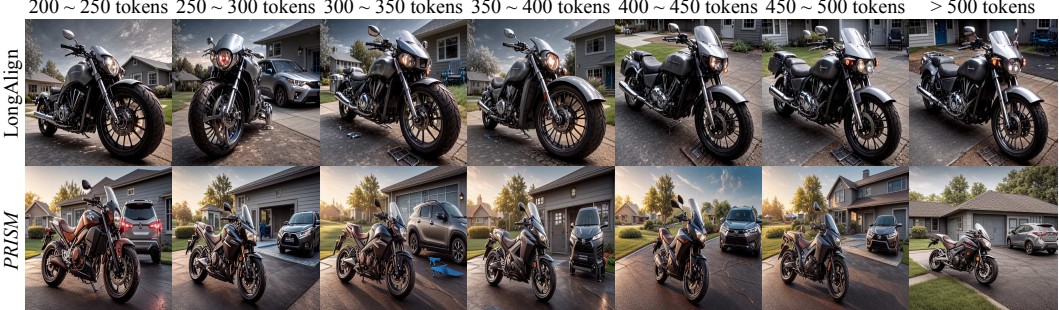

Figure 5: **Generalization to Increased Prompt Lengths.** Model fine-tuning methods (triangle mark) are effective within the training lengths but struggle to generalize to longer prompts. Projection-based methods (round mark) induce an information bottleneck and thus compromise fidelity. Leveraging compositional generalization, PRISM (square mark) maintains robust performance as input prompt lengths increase.

… The motorcycle is facing a lawn area on the side of a house … The background features a residential setting with a gray house, a lawn … The gray Toyota C-HR SUV is located to the left of the black Yamaha Virago motorcycle …

Figure 6: **Compositional Generalization.** PRISM achieves higher semantic fidelity by distributing information into multiple components. Here we sample images from a paragraph rewritten into different lengths. Compared to direct fine-tuning (LongAlign), PRISM can continuously incorporate more details as inputs expanded.

**Image Generation Quality.** We employ preference models to assess the generated image quality. We choose three CLIP-based models (CLIPScore (Hessel et al., 2021), DenScore (Bai et al., 2024), PickScore (Kirstain et al., 2023)) to evaluate overall text-image alignment, as well as MLLMs (VQAScore (Lin et al., 2024), HPSv3 (Ma et al., 2025)) for finer detail analysis. Quantitative comparisons are presented in Table 2. On a SD-1.5 backbone, PRISM-SD1.5 matches the quality of the reward-tuning model LongAlign while consistently surpassing ELLA. Crucially, our approach extends this advantage to SOTA models. PRISM-Qwen achieves the best results among modern baselines on DenScore (**22.93**), PickScore (**22.04**), and VQAScore (**86.21**), outperforming strong baselines including SD-3.5 and FLUX-Dev. Notably, our compositional strategy drastically improves the base Qwen-Image model on **HPSv3 from 8.56 to 12.05**, confirming that our framework effectively resolves the capacity bottleneck in modern foundation models to enhance long-prompt adherence. We present quantitative comparisons in Figure 4 & 7. PRISM-Qwen successfully interprets the intricate relationships and attributes among six teddy bears while other SOTA models struggles.

## 4.3 IMPROVED GENERALIZATION TO LONGER PROMPTS

T2I models are known to generalize poorly with long-prompts because of their scarcity in training data. To evaluate such generalization, we analyze the compared long-text-to-image models' performance according to input prompt length. Specifically, we partition test prompts in DetailMaster into five bins: <200 tokens, 200–300, 300–400, 400–500 and >500 tokens. As shown in Figure 5, LongAlign performs well on prompts under 300 tokens, which constitute the majority of its training data. However, its performance degrades sharply on longer prompts, dropping by up to 30% for those over 500 tokens. Although this degradation is mitigated in projection-based methods, their capacities are constrained by the fixed context window. In contrast, PRISM maintains robust performance across all prompt lengths despite being trained on the same dataset, achieving an average improvement of **7.4%** on prompts exceeding 500 tokens. This result highlights the improved generalization endowed by compositional generative modeling.

Figure 6 provides a visualization of this improved generalization. We progressively expand a base prompt with more details and compare the generated images. As prompt lengthens, elements such as the house and the yard gradually vanish in LongAlign's outputs. Our model, however, successfully integrates the additional details without overwriting existing concepts, consistently rendering all the key elements regardless of prompt length.

| StableDiffusion-3.5 | FLUX-dev. | Qwen-Image | *PRISM-Qwen* | *PRISM-FLUX.2* |

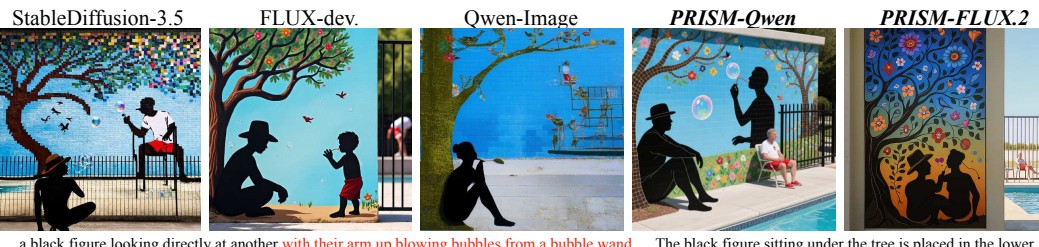

… a black figure looking directly at another with their arm up blowing bubbles from a bubble wand … The black figure sitting under the tree is placed in the lower left part of the image with a fedora style hat on with their elbows resting on their bent knees … multicolored flowers and birds in various areas of the branches … The light source appears to be front-lit, as the details of the mosaic and the lifeguard are clearly visible … To the right of the painting is a black metal fence, in the fenced area is a concrete area with a pool of water in the middle, a life guard in a white tee shirt and red shorts is present on an elevated seat …

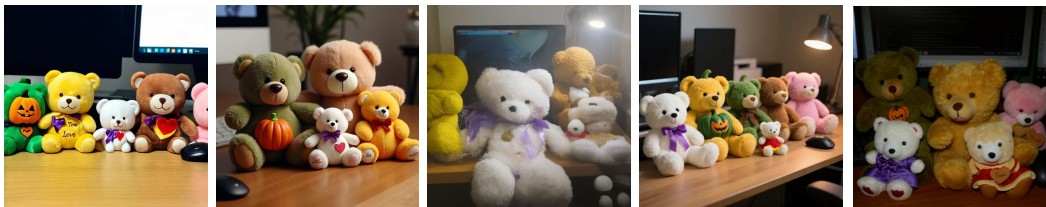

… a yellow color teddy bear with a small teddy bear in front of him … To the left of the yellow teddy bear, there is a green teddy bear with a pumpkin design … Next to the green teddy bear, there is a brown teddy bear. On the right side, we can also see a pink color teddy bear … In the lower left part of the image, there is a white teddy bear with a purple ribbon … In the lower right part of the image, there is a small white teddy bear with a red and yellow dress … In the background, we can see a computer desktop and a black mouse beside it, suggesting an indoor setting, likely a workspace or home office …

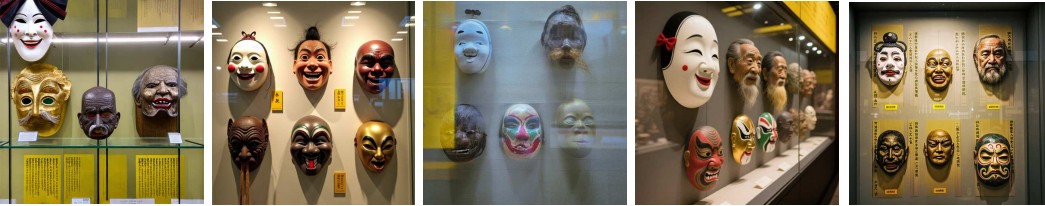

… six masks of different types, while lights and signs reflect off the glass that covers it … The first mask on the top left is a white Japanese traditional mask with a small black and red bow on the top of its head, thin black eyebrows, two small red dots on the cheeks, and curvy red lips in a smile … upper right part of the image, is of an old man with dark, rough skin, long scraggly eyebrows and a blonde mustache … in the middle, in the lower part of the image, is a vintage Chinese Beijing Opera mask … gold with an open mouth that has been twisted at the bottom to create a large, thin lip …

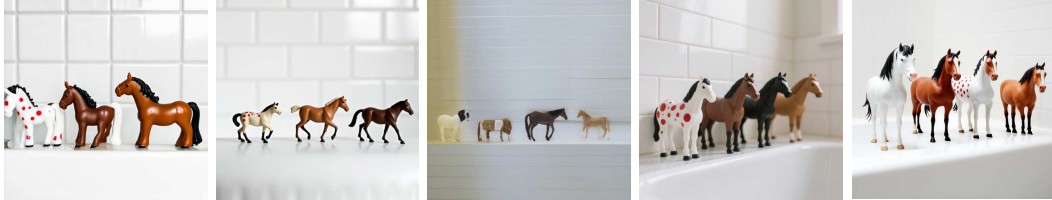

… An indoor, close up shot of the side of 4 small horse toy figures placed on the side of the bathtub, with a white tile wall directly behind the horses … The left most horse is a small white horse with a black mane and tail … The horse all the way on the right is a light brown horse with a black mane and tail, facing to the right in the right part of the image … All the horses are facing to the right … The small white horse with a black mane and tail is positioned to the left of the brown horse with a white and red dotted left half … The light brown horse with a black mane and tail is positioned to the right of the dark brown horse …

Figure 7: **Qualitative Comparisons with State-of-the-Art Baselines.** Desptie the integration of powerful LLM as text encoders, SOTA T2I models fail to capture every details in a descriptive paragraph. Our PRISM is a unversal framework that can also enhance these models' performance on out-of-distribution long prompts, allowing them to incorporate more details and render the intricate scenes.

## 4.4 ABLATION STUDY

**Number of Components.** We investigate the impact of decomposing granularity to better understand the composition benefits. Specifically, we train two additional PRISM-SD1.5 with $N = 3$ and $N = 5$ alongside our primary version with $N = 4$. Figure 9 demonstrates performance gains of these PRISM models (solid bars) over their baselines of models fine-tuned with equivalent computations. The performance gain increases with larger $N$, indicating that a finer grained decomposition has the potential to enhance compositional generalization. We further visualize this trend by generating individual outputs for each decomposed components in Figure 8. A fine grained decomposition with 5 components leads to diverse contents in individual generations, suggesting a reduced semantic load across components. Conversely, using only 3 components makes individual generations similar to their composition. This is corroborated by their similarity scores with the long-prompt encoding, that a repeated pattern is observed in the first and third columns of the similarity matrix for $N = 3$.

**Compositionality.** To further isolate the benefit from composition, we also derive non-compositional PRISM with a single component in decomposition. This baseline thus corresponds to a projection-

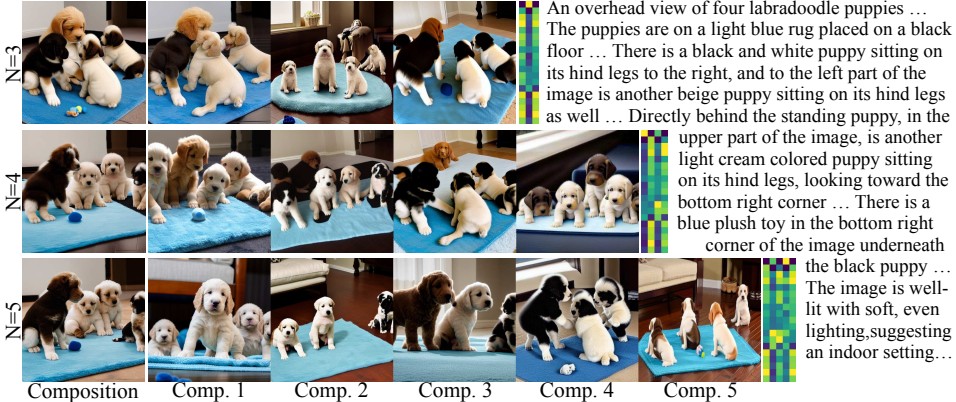

An overhead view of four labradoodle puppies … The puppies are on a light blue rug placed on a black floor … There is a black and white puppy sitting on its hind legs to the right, and to the left part of the image is another beige puppy sitting on its hind legs as well … Directly behind the standing puppy, in the upper part of the image, is another light cream colored puppy sitting on its hind legs, looking toward the bottom right corner … There is a blue plush toy in the bottom right corner of the image underneath the black puppy … The image is well-lit with soft, even lighting,suggesting an indoor setting…

Composition    Comp. 1    Comp. 2    Comp. 3    Comp. 4    Comp. 5

Figure 8: **Semantic Decoupling in Finer Grained Decomposition.** We visualize individual generation results using different numbers (N) of decomposed components. A smaller N requires each component to encode more information, causing semantic coupling. While a large N allows each component to focus on different aspects.

based long-text-to-image generation model. We increase the learnable vector's length accordingly to match the computational budgets of these variants.

As presented in Table 3, the non-compositional PRISM has a degraded performance compared to its compositional counterpart. This variant can be viewed as using the coarsest grained decomposition, thus taking the advantage of compositional generalization as discussed in Section 4.3. We also compare the performance of compositional generations using simple sentence splitting. The performance is significantly degraded due to the linguistic splitting the input paragraph loses global information in each component, resulting in inconsistent and even broken scene generations.

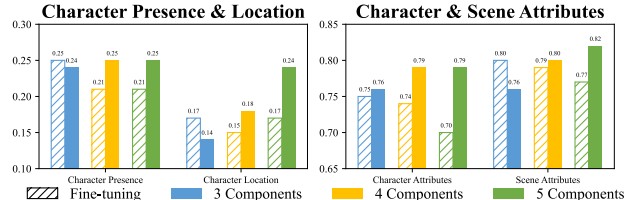

Figure 9: **Finer Grained Decomposition Improve Generalization.** More components lead to better generalization in compositional generation (solid bars) over vanilla fine-tuning (hatched bars).

Table 3: **Ablation Study.** PRISM has a degraded performance without composition. We also show the necessity of a learned decomposition by comparing with composition via sentence splitting.

| | Character Presence | Character Attributes | Character Location | Scene Attributes | Spatial Relation |
|---|---|---|---|---|---|
| Sentence Splitting | 14.01 | 72.48 | 6.44 | 58.69 | 5.18 |
| w/o Composition | 28.98 | **83.63** | 16.16 | 78.92 | 20.97 |
| w/ Composition | **29.49** | 82.97 | **17.10** | **85.34** | **22.22** |

## 5 CONCLUSION

In this paper, we address the long prompts generalization problem in T2I models. This fundamental challenge originates from the scarcity of long-captioned images in training data, which hinders T2I models from learning to render the complex narrative flow of a descriptive paragraph.

We propose PRISM, a compositional approach that leverages pre-trained T2I models' expertise on concise prompts to process out-of-distribution long prompts. PRISM uses a lightweight decomposition module to extract constituent representations from long-prompt encodings. This module is trained in an unsupervised manner, guided solely by the frozen T2I model. By distributing the rich semantic load across multiple components, PRISM demonstrates superior adherence to detailed descriptions and enhanced generalization to increased prompt lengths. Empirically, PRISM surpasses other long-text-to-image generation methods' performance by 7.4% on average. Our PRISM is also applicable to latest T2I architectures using LLM text encoders, with non-trivial improvements on the Qwen-Image model and outperforms other SOTA baselines on the generation quality.

The primary limitation of our method lies in the composition operation lacking explicit spatial control over the generation process. As a result, our method remains data-driven for factorizing the generative distributions. Future work could explore more advanced composition approaches. Another promising direction is to decompose the input prompts adaptively according to their complexity. Although we find a fixed decomposition granularity is robust as well on normal-length prompts, using fewer components for concise prompts could improve the efficiency of compositional generation.

LARGE LANGUAGE MODELS USAGE DISCLOSURE

LLMs were employed in a limited capacity for writing optimization. Specifically, the authors provided their own draft text to the LLM, which in turn suggested improvements such as corrections of grammatical errors, clearer phrasing, and removal of non-academic expressions. LLMs were also used to inspire possible titles for the paper. While the system provided suggestions, the final title was decided and refined by the authors and is not directly taken from any single LLM output. In addition, LLMs were used as coding assistants during the implementation phase. They provided code completion and debugging suggestions, but all final implementations, experimental design, and validation were carried out and verified by the authors. Importantly, LLMs were NOT used for generating research ideas, designing experiments, or searching and reviewing related work. All conceptual contributions and experimental designs were fully conceived and executed by the authors.

ETHICS STATEMENT

This research was conducted in adherence to the ICLR 2026 Code of Ethics. We specifically address the following ethical considerations:

- **Data Usage:** Our work utilizes publicly available datasets that have undergone anonymization to protect individual privacy. We have handled all data in accordance with their specified terms of use.

- **Model Bias:** Our method builds upon existing open-source Text-to-Image models. We acknowledge that these foundational models may reflect societal biases present in their training data. While a full audit of these biases is beyond the scope of our work, we highlight the importance of downstream evaluation for fairness before any real-world application of our method.

- **Societal Impact:** We recognize that Text-to-Image technology has the potential for misuse, such as the generation of misinformation. The aim of our research is to contribute positively to creative applications. We advocate for the responsible development of generative models and support community-wide efforts to establish safeguards against potential harms.

REPRODUCIBILITY STATEMENT

To ensure the reproducibility of our results, we provide our source code of the implementation of our proposed method in the supplementary material. All critical hyperparameters, training configurations and datasets details for our models can be found in Section 4.1. The computational infrastructure used for our experiments is also detailed in this section.

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

## A  IMPLEMENTATION DETAILS

### A.1  DECOMPOSITION MODULE DESIGN

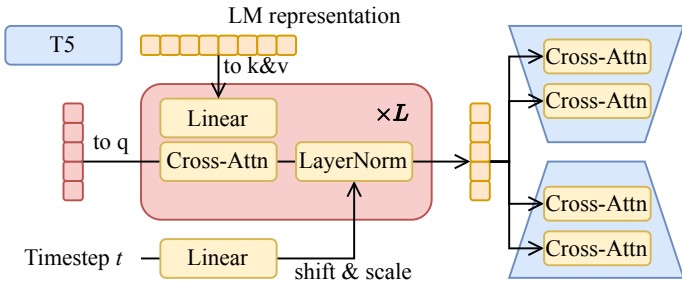

Figure 10: **Model Details for PRISM-SD1.5.** We build the decomposition module using the module design of ELLA (Hu et al., 2024), which invovles a learnable vector to query the long-prompt representation from a LM (T5) through L transformer blocks. The final output is then used as the conditional input to the pre-trained T2I model.

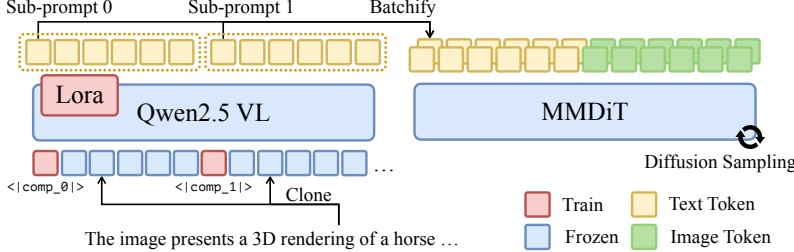

Figure 11: **Model Details for PRISM-Qwen.** We leverage the powerful text encoders in modern T2I architecture by applying LoRA and tune their text encoders to directly output constituent representations using Equation 5.

We borrow the efficient module design from ELLA (Hu et al., 2024), which contains a series of transformer blocks with a learnable query and the LM-encoded long-prompt as key-value. This architecture can efficiently extract textual condition for pre-trained T2I model from the intricate LM output. Furthermore, ELLA also introduces a time-aware adaptive layer normalization layer. This component leverages the diffusion timestep to modulate the hidden features within each transformer block, as illustrated in Figure 10. The temporal information facilitates the model to extract fine-grained textual conditions that are specific to different stages of the denoising process. The final output vector from these blocks is then sent to the cross-attention layers in T2I model, serving as the textual condition. We inherit most of their design in our PRISM-SD1.5, except that we remove the time-aware layer normalization on the query inputs which we found leads to mode collapse in the learnable vectors. PRISM-SD1.5 contains 6 transformer blocks and adopts 64 tokens in each of the N learnable queries.

For PRISM-SD3.5, we add an additional cross-attention layer in each transformer block. This additional layer accommodate the extra inputs to handle the multi text encoders in StableDiffusion-3.5 (Esser et al., 2024). We only use 3 transformer blocks to balance the overall parameter count in PRISM-SD3.5, and 128 tokens for each learnable query.

### A.2  PRISM WITH MODEL TUNING

Since the decomposed constituent representations remain in the pre-trained T2I model's expected input domain. Our PRISM can be integrated with other tuning methods efficiently. Using reward models for tuning T2I models have been widely explored recently (Kirstain et al., 2023; Wu et al., 2023; Bai et al., 2024). These models are trained on collected human preference data, and are able to measure how well the input image is aligned with text description as well as human aesthetic. We adopt the reward tuning model from LongAlign (Bai et al., 2024), which is optimized on long-caption data to provide more holistic reward signal. We apply the reward tuning algorithm from Clark et al.

(2023), which uses gradient-checkpointing to back-propagate the reward signal calculated on the final generation result:

$$\mathcal{L}(\boldsymbol{\theta}) = \mathbb{E}_{\boldsymbol{x}_0}\left[1 - \mathcal{R}(\boldsymbol{x}_0, \boldsymbol{C})\right] = \mathbb{E}_{\boldsymbol{x}_0}\left[1 - \mathcal{C}_{image}(\boldsymbol{x}_0) \cdot \mathcal{C}_{text}^T(\boldsymbol{C})\right],\tag{6}$$

where $\boldsymbol{x}_0$ is generated from our compositional long-text-to-image model using a DDIM sampler (Song et al., 2020a). We only experiment reward tuning on PRISM-SD1.5 due to our limited computation resources. For computational efficiency, we generate images with 50 sampling steps in the training loop, where we randomly choose 5 steps to calculate gradients and update model parameters within the memory constraint of our device.

### A.3 INFERENCE EFFICIENCY

Here we estimate the computational overhead of our PRISM. Theoretically, the default 4-component setting implies a $4\times$ increase in total Floating Point Operations (FLOPs). However, practical inference latency does not scale linearly with FLOPs due to hardware parallelism.

By implementing the compositional generation as a batch operation within the denoising loop, we utilize the GPU's parallel processing capabilities more effectively. This larger batch size saturates the tensor cores, mitigating the cost of the additional components. Consequently, the actual inference time increases by a factor of roughly $2\times$, rather than the theoretical $4\times$. On our hardware with A100 GPU, our method runs at 10 iterations/second, compared to LongAlign's speed of 22 iterations/second.

## B EVALUATION ON STANDARD T2I BENCHMARK

To verify that our PRISM effectively handles prompt with standard length, we evaluated its performance on standard T2I benchmarks T2I-CompBench and GenEval. As presented in Table 4, our method demonstrates robust capability in fundamental generation tasks. Specifically, we achieve the best performance on T2I-CompBench, securing the best results in the 'color', 'shape', and 'texture' metrics (ranking first in 3 out of 5 categories). Furthermore, on the GenEval benchmark, our approach remains highly competitive, achieving the second-best result with only a marginal performance difference compared to the leading baseline, LongAlign. These results confirm that our method enhances long-prompt generation capabilities without compromising fidelity or semantic alignment in standard text-to-image tasks.

Table 4: Standard T2I benchmark results on T2I-CompBench and GenEval.

| Models | Color | Shape | Texture | Spatial | Numeracy | GenEval |
|---|---|---|---|---|---|---|
| StableDiffusion-1.5 | 0.3647 | 0.3768 | 0.4095 | 0.5064 | 0.3197 | 0.4418 |
| LLM4Gen | 0.5084 | 0.4167 | 0.5085 | **0.6254** | **0.3828** | 0.4083 |
| ELLA | 0.6269 | 0.4250 | 0.5585 | 0.5713 | 0.3013 | 0.4971 |
| LongAlign | 0.5654 | 0.4693 | 0.5259 | 0.5698 | 0.3683 | **0.5075** |
| *PRISM* | **0.7113** | **0.5204** | **0.6253** | 0.6015 | 0.3701 | 0.4960 |

## C ADDITIONAL RESULTS

As demonstrated in the Table 5, the LoRA-adapted PRISM improve their baseline across the Detail-Master benchmark except the some of the scene attribute metrics. This is likely due to the chunk operation on T5 output, which may pose a risk to global information retention Similarly, we observe slight performance degradation of PRISM on the FLUX model. We also attribute this to the chunking operation on T5 outputs. We hypothesize that these limitations can be addressed through advanced module designs or by scaling training resources to support our original token-resampler design (as used in SD-1.5).

In Table 6 we evaluate images generated on the DetailMaster benchmark using CLIPScore, DenScore, PickScore and HPSv3, where the best results are all obtained by LoRA-adapted PRISM except on the CLIPScore metric which is unreliable in capturing long-prompt semantics.

Table 5: DetailMaster benchmark results of **PRISM** on SD1.5, SD3.5, Qwen-Image and FLUX.

| Model | Character Presence | Character Attributes | | | Character Location | Scene Attributes | | | Spatial Relation |
|---|---|---|---|---|---|---|---|---|---|
| | | Object | Animal | Person | | Background | Light | Style | |
| StableDiffusion-1.5 | 15.26 | 24.82 | 11.48 | 11.99 | 7.39 | 22.02 | 65.81 | 83.91 | 5.75 |
| **PRISM-SD1.5** | 23.37 | 28.39 | 27.95 | 15.72 | 15.30 | 78.17 | 89.21 | 74.95 | 17.33 |
| StableDiffusion-3.5-M | 31.19 | 31.55 | 32.03 | 27.54 | 26.69 | 87.89 | 92.32 | 94.70 | 28.90 |
| *PRISM-SD3.5* | 33.03 | 30.28 | 35.37 | 31.21 | 27.62 | 87.16 | 91.96 | 92.14 | 31.98 |
| Qwen-Image | 31.63 | **41.01** | 36.22 | 24.15 | 31.01 | 92.29 | **96.34** | 91.41 | 37.14 |
| *PRISM-Qwen* | 36.84 | 38.17 | **40.50** | 29.95 | **32.55** | 85.87 | 95.43 | **94.70** | **37.80** |
| FLUX-Dev. | 34.33 | 38.49 | 38.40 | **32.30** | 31.62 | **92.84** | 95.80 | 94.70 | 35.31 |
| *PRISM-FLUX* | 31.05 | 36.70 | 34.46 | 27.83 | 26.69 | 85.69 | 93.24 | 91.96 | 30.29 |

Table 6: Quantitative evaluations of image generation quality on large-scale T2I models.

| Models | CLIPScore | DenScore | PickScore | HPSv3 |
|---|---|---|---|---|
| StableDiffusion-3.5 Medium | 34.97 | 22.37 | 21.63 | 13.39 |
| *PRISM-SD3.5* | 32.97 | **25.01** | 21.49 | **13.52** |
| Qwen-Image | 33.85 | 22.25 | 20.98 | 8.556 |
| *PRISM-Qwen* | **34.12** | **22.93** | **22.04** | **12.05** |
| FLUX-Dev. | 33.30 | 22.56 | 21.89 | 13.17 |
| *PRISM-FLUX* | 32.10 | 22.36 | 21.63 | 12.78 |

## D  FULL TEXT PROMPTS FOR IMAGE GENERATION

In this section we provide the full long-prompt that is used for generating figures in this paper.

For generating Figure 1:

1. The image presents a 3D rendering of a horse, captured in a profile view. The horse is depicted in a state of motion, with its mane and tail flowing behind it. The horse's body is composed of a network of lines and curves, suggesting a complex mechanical structure. This intricate design is further emphasized by the presence of gears and other mechanical components, which are integrated into the horse's body. The background of the image is a dark blue, providing a stark contrast to the horse and its mechanical components. The overall composition of the image suggests a blend of organic and mechanical elements, creating a unique and intriguing visual.

2. A hyper-detailed, macro shot of a human eye, presented not as an organ of sight, but as a gateway to a lost world of intricate craftsmanship. The iris is a masterfully crafted, antique horological mechanism, a complex universe of miniature, interlocking gears and cogs made from polished brass, copper, and tarnished silver. Each metallic piece is exquisitely detailed, with tiny, functional teeth that seem to pulse with a slow, rhythmic, and almost imperceptible life. The vibrant color of the iris is replaced by the warm, metallic sheen of the gears, with ruby and sapphire jewels embedded as tiny, gleaming pivots. At the center, the pupil is not a void but the deep, dark face of a miniature clock, its impossibly thin, filigreed hands frozen at a moment of profound significance. The delicate, thread-like veins in the sclera are reimagined as fine, coiling copper wires, connecting the central mechanism to the unseen power source at the edge of the frame. The entire piece is captured under a soft, focused light that highlights the metallic textures and casts deep, dramatic shadows within the complex machinery, suggesting immense depth. The background is a stark, velvety black, ensuring nothing distracts from the mesmerizing, mechanical soul of the eye.

3. A sleek, enigmatic feline, a cat of indeterminate breed, is the central figure, poised in a state of serene contemplation. Its body is not of flesh and bone, but meticulously sculpted from a complex lattice of polished, interlocking obsidian shards. Each piece is perfectly fitted against the next, creating a mosaic of deep, lustrous black that absorbs the light. The cat's form is defined by the sharp, clean edges of these volcanic glass fragments, giving its natural curves a subtle, geometric undertone. Glimmering veins of molten gold run through the cracks between the shards, glowing with a soft, internal heat that pulses rhythmically, like a slow heartbeat. These golden rivers trace the contours of the cat's muscles and skeleton, outlining its elegant spine, the delicate structure of its paws, and the graceful curve of its tail. Its eyes are two brilliant, round-cut rubies, catching an unseen light source and

Table 7: Image Quality Assessment on LongAlign dataset.

| Metrics | SD-1.5 | LLM4GEN | ELLA | LongAlign | *PRISM* |
|---|---|---|---|---|---|
| CLIPScore | 0.3462 | 0.3362 | 0.3310 | 0.3568 | 0.3519 |
| DenScore | 0.2047 | 0.2028 | 0.2112 | 0.2587 | 0.2596 |
| PickScore | 0.2083 | 0.2069 | 0.2052 | 0.2306 | 0.2308 |
| HPSv3 | 9.174 | 7.620 | 5.631 | 12.72 | 12.61 |

casting a faint, crimson glow. The whiskers are impossibly thin strands of spun platinum, fanning out from its muzzle with metallic precision. The entire figure rests upon a simple, unadorned, and dimly lit surface, ensuring that all focus remains on the cat's extraordinary construction—a masterful fusion of natural grace and exquisite, dark craftsmanship.

For generating Figure 4:

1. A high angle shot of a brown wooden bench with several dishes on top of it. In the center and on the left are two round, wavy side plates with black scratches on the sides and a doily pattern engraved on the plates. On both plates is a thick brown cookie that's been crosscut at the top, located in the middle part of the image. The plate on the right has a candy with a yellow wrapper and green ends. To the right of the plates is a white mug with whipped cream on top that is similar to the glass plates. The cup, made of ceramic material, has a cylindrical shape with a handle and a textured surface. The white whipped cream on top is frothy and has an embossed design. Surrounding the wooden bench is a dark brown wooden floor. On the top right is a gray curtain, and on the upper left is a view of the lower part of a white wooden wall. The image is taken indoors with soft, warm lighting, likely from an overhead source, creating a cozy and inviting atmosphere. The lighting is evenly distributed, with no harsh shadows, suggesting a relaxed time of day, possibly evening. The style of the image is a realistic photo with a warm, homely aesthetic. The brown wooden bench supports the two round, wavy side plates with black scratches and a doily pattern, which are placed side by side. The thick brown cookies crosscut at the top are positioned on top of the two round, wavy side plates, with one cookie on each plate. The candy with a yellow wrapper and green ends is located on the right plate, next to the thick brown cookie. The white mug with whipped cream on top is situated to the right of the two round, wavy side plates. The two round, wavy side plates are adjacent to each other, with the plate containing the candy being closer to the white mug with whipped cream on top.

2. An indoor top-down view of a wooden Statue of Liberty, which is positioned centrally on the table, covering a black marking on the table, on a wooden table with 3 wooden cars and 1 wooden limo next to it. The wooden limo is placed to the left of the wooden Statue of Liberty, and the three wooden cars are arranged to the right of the wooden Statue of Liberty. On the table, the black marking on the table is partially hidden by the wooden Statue of Liberty in the upper part of the image. Behind the table is a dark blue curtain, through which sunlight is coming and shining down on the right side of the table, casting a soft glow and creating gentle shadows that highlight the wooden textures. The lighting is soft and natural, suggesting it is daytime with sunlight filtering through the curtain, illuminating the right side of the table. The dark blue curtain is located behind the table, indicating it is not on the same plane as the objects on the table, and it is positioned at the back of the image. The style of the image is a realistic photo.

3. A long-shot view of a slightly dark sky with a cumulonimbus forming in the clouds, allowing rays of sunlight to pierce through, creating a striking contrast against the darkened landscape. The sky is bright blue, and the cumulonimbus cloud formation is a dark blue and gray, with soft, diffused sunlight breaking through the clouds, suggesting it is either early morning or late afternoon, with the sun low on the horizon. A small house is visible in the distance; it has tan panels, and it has a white metal roof. Parked in the lower right part of the image in front of the house is a white sedan, situated between the house and the viewer. Surrounding the house are many tall, healthy trees that are mostly shrouded in shadow; these trees have green leaves, a broad canopy, dense foliage, and provide natural shade, located around and behind the small house, creating a natural border. The grass surrounding them is evenly cut

and healthy. The scene is somewhat dark, with rays of sunlight shining through the gathered clouds to illuminate the sky from above, enhancing the tranquil yet moody atmosphere. The cumulonimbus cloud formation is positioned above the small house, and the rays of sunlight are directed towards the area above the house and trees, capturing natural lighting and atmospheric conditions in a realistic photo style.

4. A front view of a parking lot with several vehicles parked including two dark colored sedans in the middle part of the image and what appears to be six different motor bikes in front of them. The bikes seem to range from a red motorbike with color red, material metal and plastic, typical features include two wheels, handlebars, seat, engine, exhaust pipe, and headlight in the right part of the image, a white motor bike in the left part of the image, a silver motor bike, another white motor bike, another silver motorbike that is silver in color, and another silver motorbike that is also silver in color. The parking has visible but faded white parking lines, and behind all of the vehicles are two handicap parking signs. Behind the handicap signs is a large cream colored building that covers all but the top left side of the background view, it has a partially visible blue colored roof and a red colored rectangular shaped strip that passes along the view of the building a couple of feet below the blue roof. The background features a large cream-colored building with a blue roof and a red strip, partially obscured by the parked vehicles. Two handicap parking signs are visible on the building's facade. The image appears to be taken during the day under natural light, with the light source positioned overhead, creating soft shadows beneath the vehicles. The lighting is bright and even, suggesting a clear sky with no direct sunlight causing harsh shadows. The style of the image is a realistic photo. The two dark colored sedans are positioned behind the motorbikes, with one slightly to the left and the other to the right. The red motorbike is to the right of the other motorbikes, closer to the right sedan. The white motorbike is to the far left, with the silver motorbike next to it. The another white motorbike is positioned between the first white motorbike and the silver motorbikes. The another silver motorbike is next to the another white motorbike, and the last silver motorbike is next to the red motorbike. The cream colored building with a blue roof and red strip is behind all the vehicles, with the handicap signs in front of it.

For generating Figure 6 and Figure 8:

1. A high-angle side view of a black Yamaha Virago motorcycle facing the right side of the image parked on an black asphalt surface. The front of the motorcycle is turned slightly toward the top right corner of the image. The fenders, the fuel tank, and the handles of the motorcycle are black. The motorcycle has a brown leather seat. The engine, exhaust pipes, and handlebar are gray silver. There is a red tail light attached to the fender over the top of the rear wheel. The Virago logo is on the side of the gas tank. The motorcycle is facing a lawn area on the side of a house visible at the top of the image. There is a patch of grass and a walkway leading to a gray door near the top right corner of the image, there is a window on each side of the door. There are two blue chairs in the top right corner of the image. Visible in the top left corner of the image is the right side of the front of a gray Toyota C-HR SUV with metallic paint, a compact SUV shape, sleek headlights, a Toyota emblem, and a modern design. The background features a residential setting with a gray house, a lawn, a walkway, and two blue chairs near the top right corner. A gray Toyota C-HR SUV is partially visible in the top left corner. The image is taken outdoors under natural daylight, with soft lighting conditions suggesting it could be morning or late afternoon. The light source is positioned to the side, creating gentle shadows and highlighting the motorcycle's details. The style of the image is a realistic photo. The black Yamaha Virago motorcycle is positioned in front of the lawn area with a gray door and windows, indicating it is closer to the viewer than the house. The gray Toyota C-HR SUV is located to the left of the black Yamaha Virago motorcycle, suggesting it is parked parallel to the motorcycle but further away from the house. The two blue chairs are situated to the right of the lawn area with a gray door and windows, showing they are placed on the side of the house away from the motorcycle and the SUV. The lawn area with a gray door and windows is between the motorcycle and the two blue chairs, establishing it as a central point in the spatial arrangement of the scene.

2. An overhead view of four labradoodle puppies, three puppies are sitting and one puppy is standing with its right paw resting against the white barrier at the bottom of the image. The puppies are on a light blue rug placed on a black floor. The puppy standing is a beige

and white puppy with curly fur, dark eyes, a small nose, and a fluffy appearance, its paw extended. There is a black and white puppy sitting on its hind legs to the right, and to the left part of the image is another beige puppy sitting on its hind legs as well. Directly behind the standing puppy, in the upper part of the image, is another light cream colored puppy sitting on its hind legs, looking toward the bottom right corner of the image. The three puppies in the front are looking up, the puppy behind them is looking toward the bottom right corner of the image. There is a blue plush toy in the bottom right corner of the image underneath the black puppy. The rug the puppies are on is not laying completely flat on the ground, its unintentionally folded up in some areas and folded over itself in the top right corner of the image. The background consists of a light blue rug placed on a black floor, with the rug showing some unintentional folds and overlaps. A blue plush toy is visible in the bottom right corner under the black puppy. The image is well-lit with soft, even lighting, suggesting an indoor setting with artificial light sources. The light appears to be front-lit, as there are no harsh shadows on the puppies. The style of the image is a realistic photo. The beige and white puppy standing with its right paw resting against the white barrier is in front of the light cream colored puppy sitting on its hind legs in the back. The black and white puppy sitting on its hind legs to the right is to the right of the beige and white puppy standing with its right paw resting against the white barrier. The beige puppy sitting on its hind legs to the left is to the left of the beige and white puppy standing with its right paw resting against the white barrier. The light cream colored puppy sitting on its hind legs in the back is behind the beige and white puppy standing with its right paw resting against the white barrier. The black and white puppy sitting on its hind legs to the right is next to the beige puppy sitting on its hind legs to the left.

