# OpenReview forum: "Long-Text-to-Image Generation via Compositional Prompt Decomposition"
_ICLR.cc/2026/Conference — ICLR 2026 Poster_

### Official Review · Reviewer_rbyh · 2025-10-26

**Soundness:** 2
**Presentation:** 3
**Contribution:** 2
**Rating:** 4
**Confidence:** 4

**Summary:**

This paper addresses the limitation of modern text-to-image models in capturing fine-grained details from long text inputs. The authors propose a trainable PromptDecomposer module that decomposes lengthy text prompts into multiple semantically coherent sub-prompts. Experimental results demonstrate that the proposed method achieves strong performance on the challenging DetailMaster benchmark.

**Strengths:**

A key strength of this work lies in the latent-space prompt decomposition strategy. By decomposing prompts in the latent space, the method avoids the semantic fragmentation issues often observed when splitting raw text. This leads to more coherent sub-prompt representations and enables the model to balance local detail preservation with global semantic consistency, contributing to improved text-image alignment.

**Weaknesses:**

1. Limited Adaptability to Different Backbones:
As acknowledged by the authors, the proposed method exhibits scalability issues when applied to larger diffusion backbones. Specifically, both the training and inference costs grow significantly with model size, which raises concerns about its practicality on modern large-scale models such as SD3, FLUX, SD3.5, or Qwen-Image. This limitation restricts the method’s usability in real-world or production-level scenarios.

2. Insufficient Experimental Coverage:
To ensure fair and comprehensive evaluation, the method should also be tested on the Evaluation Dataset proposed in LongAlign [1]. Without such comparison, it is difficult to judge whether the performance gains are consistent across different long-text benchmarks.

3. Lack of Verification on Short Prompts:
The paper focuses primarily on long-text generation, but does not evaluate whether the proposed approach compromises the model’s ability to handle short prompts. Experiments on standard short-prompt benchmarks such as GenEval [2] and T2I-CompBench++ [3] are necessary to demonstrate the generalization and robustness of the proposed method.

4. Missing Inference Efficiency Comparison:
A comparison of inference memory consumption and latency between this method and LongAlign [1] would help clarify the trade-offs between alignment improvement and computational overhead, providing a more complete understanding of its practical value.

[1] Improving Long-Text Alignment for Text-to-Image Diffusion Models.

[2] GenEval: An Object-Focused Framework for Evaluating Text-to-Image Alignment.

[3] T2I-CompBench++: An Enhanced and Comprehensive Benchmark for Compositional Text-to-image Generation.

**Questions:**

1. The training data used in this paper differs from that used in LongAlign [1], which may lead to unfair comparison and make it difficult to attribute performance improvements solely to the proposed method.

2. The PromptDecomposer module appears to have limited contribution when considering the scaling behavior across different model backbones. How does the proposed framework adapt or generalize to models with varying capacities?

3. In [2], a training-free method for processing long texts at the sentence-level can be added to the baseline for reference, which can make the experiment more comprehensive.

[1] Improving Long-Text Alignment for Text-to-Image Diffusion Models.

[2] Hybrid Layout Control for Diffusion Transformer: Fewer Annotations, Superior Aesthetics.

---

> ### Author Response · Authors · 2025-11-21
> **Official Response to Reviewer rbyh (1)**
>
> We sincerely thank the reviewer for the positive and encouraging feedback. We appreciate the recognition of the latent space decomposition strategy that facilitates global semantic coherence of our method. We address the reviewer’s concerns in detail below.
>
> # [Q2] Transferability to modern architecture
> > Limited Adaptability to Different Backbones: As acknowledged by the authors, the proposed method exhibits scalability issues when applied to larger diffusion backbones. Specifically, both the training and inference costs grow significantly with model size, which raises concerns about its practicality on modern large-scale models such as SD3, FLUX, SD3.5, or Qwen-Image. This limitation restricts the method’s usability in real-world or production-level scenarios.
>
> > How does the proposed framework adapt or generalize to models with varying capacities?
>
> We sincerely appreciate the reviewer for raising the concern on transferring our method to a wider range of model architectures. Our framework can be adapted to modern architectures (eg., SD3.5 and Qwen-Image) with module design optimization. While simply scaling the module design for a SD-1.5 backbone to these large-capacity models exceeds our computational resources, we propose a workaround by leveraging **parameter-efficient LoRA-adaptation** (detailed in **Appendix C.1 and Figure 10**).
>
> - **LLM-based Text Encoders (e.g., Qwen-Image):** We leverage the reasoning capabilities of the decoder-only LLMs. By introducing learnable component-tokens and fine-tuning via LoRA, we enable the text encoder to directly output decomposed representations.
> - **T5-based Text Encoders (e.g., SD3.5, FLUX):** We apply LoRA to the text encoder and split the output into decomposed embeddings according to the number of decomposed components. For SD3.5, we let PromptDecomposer process the CLIP representation separately and concatenates it with the decomposed T5 representations to match its multi-encoder design.
>
> # [W1] Enhancing SOTA baselines
>
> > The PromptDecomposer module appears to have limited contribution when considering the scaling behavior across different model backbones.
>
> We thank the reviewer for raising the critical question regarding scalability to modern architectures. As discussed in our response for [Q2], simply scaling the PromptDecomposer design for SD-1.5 can be computationally demanding for larger backbones. To address this, we introduced a parameter-efficient **LoRA-adapt PromptDecomposer** as detailed in **Appendix C.1 and Figure 10**.
> 1. **Resolving Scalability (10x Reduction):** Instead of scaling up the standalone transformer module, we apply LoRA to fine-tune the text encoders in modern architecture to directly output decomposed representations. This strategy drastically reduces the training overhead, requiring only **160M trainable parameters**—a **10× reduction** compared to the naive transfer—making the SOTA models trainable on our device.
>
> - **Benchmark Evaluations:** As shown in the table below, our method effectively improves **Qwen-Image** and **SD-3.5** representing the modern baselines on the DetailMaster Benchmark. On Qwen-Image that employs Qwen2.5-VL as text-encoder, our methods improves **Object Presence by +5.2%** and **Character Attributes by 3.69%**. On SD3.5, the LoRA-adapted PromptDecomposer also improves the vanilla SD-3.5 with **+3.08% in Spatial Relation and +1.84% in Object Presence**.
>
> | Models | object presence | character attributes | character location | scene attributes | spatial |
> |:---|:---:|:---:|:---:|:---:|:---:|
> | StableDiffusion-3.5 Medium | 31.19 | 30.37 | 26.69 | 91.64 | 28.90 |
> | PromptDecomposer-SD3.5 | **33.03** | **33.66** | **27.62** | 90.42 | **31.98** |
> | Qwen-Image | 31.63 | 33.89 | 31.01 | 93.35 | 37.14 |
> | PromptDecomposer-Qwen | **36.84** | **37.58** | **32.55** | 92.01 | **37.80** |
>
> - **Image Quality Assessment:** As shown in **Table 6** as well as the table below, our method improves the powerful Qwen-Image with **HPSv3 from 8.556 to 12.05** and **DenScore by approximately +12%** on SD3.5, demonstrating that our compositional approach effectively enhances the long-prompt generation performance of SOTA models. Note that CLIPScore utilizes CLIP for assessing the text-image alignment and is thus unreliable in measuring the semantics of long-captions. We also provide some quantitative comparisons in **Figure 12 & 13**, as well as a user study compared with SOTA models in **Figure 11** in the revised paper.
>
> | Models | CLIPScore | DenScore | PickScore | HPSv3 |
> | :--- | :--- | :--- | :--- | :--- |
> | StableDiffusion-3.5 Medium | 34.97 | 22.37 | 21.63 | 13.39 |
> | **PromptDecomposer-SD3.5** | 32.97 | **25.01** | 21.49 | **13.52** |
> | Qwen-Image | 33.85 | 22.25 | 20.98 | 8.556 |
> | **PromptDecomposer-Qwen** | **34.12** | **22.93** | **22.04** | **12.05** |

---

> ### Author Response · Authors · 2025-11-21
> **Official Response to Reviewer rbyh (2)**
>
> # [Q1] Training data
> > The training data used in this paper differs from that used in LongAlign [1], which may lead to unfair comparison and make it difficult to attribute performance improvements solely to the proposed method.
>
> We thank the reviewer for raising this important point regarding comparison fairness. We would like to clarify that our model was trained on the **identical dataset** released and used by LongAlign (Bai et al., 2024), as stated in Section 4.1 (Line 269). Specifically, we utilized the same ~2 million re-captioned images from SAM, COCO2017, LLaVA, and JourneyDB to train PromptDecomposer-SD1.5. By strictly controlling for the training data, we ensure that the reported performance improvements are attributable to our compositional architecture rather than data discrepancies.
>
> # [W2] Insufficient evaluation
> > Insufficient Experimental Coverage: To ensure fair and comprehensive evaluation, the method should also be tested on the Evaluation Dataset proposed in LongAlign [1]. Without such comparison, it is difficult to judge whether the performance gains are consistent across different long-text benchmarks.
>
> We appreciate the reviewer’s suggestion to ensure a more comprehensive evaluation. To address this, we evaluated our method on the dataset proposed in LongAlign [1] using their specified configuration. We report the CLIPScore, DenScore, PickScore and HPSv3 in the new **Table 7** added to the Appendix.
>
> As shown below, **PromptDecomposer** achieves performance comparable to **LongAlign**, while significantly outperforming other baselines (SD-1.5, LLM4GEN, and ELLA). This confirms that our compositional approach maintains high fidelity even on evaluations tailored for tuning-based methods.
>
> | Metrics | SD-1.5 | LLM4GEN | ELLA | LongAlign | PromptDecomposer
> |:---|:---:|:---:|:---:|:---:|:---:|
> | CLIPScore | 0.3462 | 0.3362 | 0.3310 | 0.3568 | 0.3519 |
> | DenScore | 0.2047 | 0.2028 | 0.2112 | 0.2587 | 0.2596 |
> | PickScore | 0.2083 | 0.2069 | 0.2052 | 0.2306 | 0.2308 |
> | HPSv3 | 9.174 | 7.620 | 5.631 | 12.72 | 12.61 |
>
> [1] Improving Long-Text Alignment for Text-to-Image Diffusion Models
>
> # [W3] Standard T2I benchmark
> > Lack of Verification on Short Prompts: The paper focuses primarily on long-text generation, but does not evaluate whether the proposed approach compromises the model’s ability to handle short prompts. Experiments on standard short-prompt benchmarks such as GenEval [2] and T2I-CompBench++ [3] are necessary to demonstrate the generalization and robustness of the proposed method.
>
> We appreciate this suggestion and agree that verifying performance on standard prompts is essential to ensure our compositional approach generalizes well. We have conducted evaluations on **T2I-CompBench and GenEval** as requested; these results are detailed in **Appendix B (Table 4)** of the revised paper.
>
> As shown in the table below, our method creates no performance regression on short prompts. In fact, **PromptDecomposer** achieves the highest scores in the 'Color', 'Shape', and 'Texture' metrics on T2I-CompBench (ranking 1st in 3 out of 5 categories). Furthermore, on the 'spatial' and 'numeracy' metrics as well as the GenEval benchmark, our method remains highly competitive, achieving the second-best result with only a marginal difference (<1.2%) compared to the top baseline. These results confirm that our decomposition strategy enhances model generalization to longer prompts without compromising the model's ability to handle standard, shorter inputs.
>
> | Models | color | shape | texture | spatial | numeracy | GenEval |
> |:---|:---:|:---:|:---:|:---:|:---:|:---:|
> | SD-1.5 | 0.3647 | 0.3768 | 0.4095 | 0.5064 | 0.3197 | 0.4418 |
> | LLM4GEN | 0.5084 | 0.4167 | 0.5085 | **0.6254** | **0.3828** | 0.4083 |
> | ELLA | 0.6269 | 0.4250 | 0.5585 | 0.5713 | 0.3013 | 0.4971 |
> | LongAlign | 0.5654 | 0.4693 | 0.5259 | 0.5698 | 0.3683 | **0.5075** |
> | PromptDecomposer | **0.7113** | **0.5204** | **0.6253** | 0.6015 | 0.3701 | 0.4960 |

---

> ### Author Response · Authors · 2025-11-21
> **Official Response to Reviewer rbyh (3)**
>
> # [W4] Discussion on Inference efficiency
> > Missing Inference Efficiency Comparison: A comparison of inference memory consumption and latency between this method and LongAlign [1] would help clarify the trade-offs between alignment improvement and computational overhead, providing a more complete understanding of its practical value.
>
> We thank the reviewer for bringing up this crucial point. For the default 4-components compositional generation, total FLOPs will be increased by 4 times theoretically. However, since we implement the compositional generation as batch operation inside the denoising loop, increasing the batch size actually "saturates" the GPU's tensor cores effectively, allowing it to process the extra data in parallel rather than sequentially. The actual inference time is increased by roughly **2 times only**. On our A100 GPU, our compositional generation runs at a speed of _10 iteration/s_ in the denoising loop, compared to LongAlign’s _22 iteration/s_. The discussion on inference efficiency has been included in *Appendix A.3 (Page 14)*.
>
> # [Q3] Training-free baseline
> > In [2], a training-free method for processing long texts at the sentence-level can be added to the baseline for reference, which can make the experiment more comprehensive.
>
> We thank the reviewer for suggesting the inclusion of training-free baselines. We have addressed this suggestion in two ways:
>
> 1. **Sentence-Level Splitting:** We evaluated a training-free method that decomposes long prompts via sentence splitting (**Table 3**, "Decomposition via Split"). As shown, this approach yields poor performance (Character Presence: 14.01%), confirming that simple linguistic splitting destroys global context and fails to impose spatial control.
>
> 2. **Training-Free Layout Generation:** Regarding specific layout-guided methods like HybridLayout, we note they often require ground-truth bounding boxes, which are unavailable in standard T2I settings. To provide a fair comparison, we instead evaluated **LLM Blueprint**, a SOTA training-free method that uses an LLM to synthesize layout conditions for downstream layout-control models. We have added these results to **Table 1**. As shown below, LLM Blueprint significantly underperforms our compositional approach, except on the 'character location' metric benefit from the explicit layout control.
>
> | Models | object presence | character attributes | character location | scene attributes | spatial |
> |:---|:---:|:---:|:---:|:---:|:---:|
> | LLM Blueprint | 18.69 | 78.06 | 18.40 | 69.01 | 14.16 |
> | PromptDecomposer-SD1.5 | **25.99** | **86.14** | 16.21 | **89.02** | **24.47** |

---

### Official Review · Reviewer_2frZ · 2025-10-31

**Soundness:** 3
**Presentation:** 3
**Contribution:** 2
**Rating:** 4
**Confidence:** 4

**Summary:**

Compositional approach for long text-to-image generation using learnable queries to decompose long-prompt representations into sub-prompts in representation space, processed in parallel through frozen pre-trained T2I models and merged via concept conjunction.
Achieves 7.4% better generalization on 500+ token prompts on SD 1.5-based baselines.

**Strengths:**

1) Interesting compositional framework grounded in energy-based models.

2) Efficient for legacy models: trains only PromptDecomposer (~20hrs on 4 A100s), avoiding large fine-tuning costs.

3) Strong generalization results. 7.4% improvement on 500+ tokens is substantial.

4) Comprehensive evaluation on DetailMaster. With thorough ablation studies and multiple metrics.

4) Clear presentation with good visualizations, honest limitation discussion.

**Weaknesses:**

1. **Questionable practical motivation with outdated baselines**: Focuses on SD 1.5 (2022) while modern models (SD 3.5, Flux) with T5-XXL encoders likely handle long prompts well.
> Authors must demonstrate that these SOTA models actually fail on long prompts to justify the compositional approach, and provide comprehensive DetailMaster benchmark comparisons showing where SD 1.5-based methods fall short relative to modern baselines. At least when I see Table 4 result, Vanilla SD 3.5 already shows pretty good result in numbers compared to PromptDecomposer+SD1.5+Tuned or all the other baselines. I cannot understand why we should use PromptDecomposer, if the use of SD 3.5 (or Flux -- I want this added as baseline comparsion too.) is already better in numbers.


2. **Poor scalability to modern architectures**: SD 3.5 experiments (Table 4) show minimal improvements despite requiring 1.2B parameters for PromptDecomposer-SD3.5; moreover, Table 4 reveals SD 3.5 already achieves strong scores (CLIPScore 34.97, DenScore 22.37) suggesting the problem may not exist for modern models. No experiments with Flux/SDXL provided. The approach appears limited to weak text encoders (CLIP), and generalization across different text-encoder architectures (CLIP vs T5 vs T5-XXL) needs systematic demonstration.
> Please add text-encoder comparison experiments for clear demonstration of your contribution.

3. **Evaluation Metrics** : In Tab. 2, you reported HPSv3, but in Tab. 4, you reported HPSv2. Is it Typo? If it is typo, I see that PromptDecomposer or other baselines in Tab. 2 stays at around 6.7 (ELLA) to 13.26 (LongAlign), but using vanilla SD 3.5 shows 28.86. Pickscore and Denscore shows better numbers in PromptDecomposer or on some baselines, but difference is not as big compared to HPS difference.
> I want more explanations on this typo and number differences. Also, please add SD 3.5 / Flux vanilla (w/o tuning or adding PromptDecomposer component) to the main table for clear comparison. Also for these SOTA models I think adding qualitative comparison might be helpful if the actual result of PromptDecomposer demonstrating the long-prompt is better than those SOTA models.

**Questions:**

**Check weaknesses section above for details.**

The major weakness of this work is the **questionable motivation** and **lack of evaluation/demonstration on state-of-the-art models**. The paper focuses on SD 1.5 (2022) while modern models with stronger text encoders (SD 3.5, Flux with T5-XXL) likely already handle long prompts effectively, yet no comprehensive comparison is provided. Table 4 shows SD 3.5 already achieves strong performance, and the minimal improvement from PromptDecomposer-SD3.5 raises fundamental questions about whether this problem still exists for current models.

**To increase my score, the authors should address:**
1. Demonstrate that SOTA models (SD 3.5, Flux) actually fail on long prompts with DetailMaster benchmark results (or at least on qualitative results.)
2. Include these models as baselines in main comparison tables
3. Provide ablations showing improvements come from decomposition rather than just better text encoding (e.g., T5-XXL with SD 1.5 without decomposition)
4. Add user studies comparing generation quality against modern models

If these additional experiments convincingly show the compositional approach provides value beyond simply using better text encoders, I will reconsider my evaluation.

---

> ### Author Response · Authors · 2025-11-21
> **Official Response to Reviewer 2frZ (1)**
>
> We sincerely thank the reviewer for the positive and encouraging feedback. We appreciate the recognition of the superior generalization results of our method based on a compositional framework in the energy-based perspective, as well as the acknowledgement of its efficiency and effectiveness demonstrated through comprehensive experiments. We address the reviewer’s concerns in detail below.
>
> # [Q1, Q2, W1] Limitations of SOTA models
> > Demonstrate that SOTA models (SD 3.5, Flux) actually fail on long prompts with DetailMaster benchmark results (or at least on qualitative results.)
>
> > Authors must demonstrate that these SOTA models actually fail on long prompts to justify the compositional approach,
>
> > Also, please add SD 3.5 / Flux vanilla (w/o tuning or adding PromptDecomposer component) to the main table for clear comparison.
>
> > Include these models as baselines in main comparison tables
>
> We appreciate the reviewer for raising the concern regarding modern baselines. We agree that verifying the limitations of SOTA models is essential to justify the need for compositional approaches. To address this, we have evaluated **SD-3.5, Qwen-Image, and FLUX** on the DetailMaster benchmark, and listed these results compared with SD-1.5 in **Table 5** of the revised PDF. The results confirm that even SOTA models struggle significantly with paragraph-length prompts:
>
> | Models | object presence | character attributes | character location | scene attributes | spatial |
> |:---|:---:|:---:|:---:|:---:|:---:|
> | SD-3.5-M | 31.19 | 30.37 | 26.69 | 91.64 | 28.90 |
> | Qwen-Image | 31.63 | 33.89 | 31.01 | 93.35 | 37.14 |
> | FLUX | 34.33 | 36.90 | 31.62 | 94.45 | 35.31 |
>
> _Note: To rigorously quantify "failure", the ‘Character Attributes’ reported here are **un-normalized** (joint accuracy), meaning they account for missing objects. If an object is not generated, it is counted as a failure for its attributes._
>
> Key Observations:
> 1. **Significant Failure Rate:** As shown above, even the strongest baseline (FLUX-Dev) achieves only **34.33%** in Character Presence. This indicates that nearly **65% of the objects** described in long prompts are completely omitted by SOTA models.
> 2. **Justification for Composition:** This high omission rate validates our core premise: SOTA models still face a "capacity bottleneck" when processing dense, long-form text, necessitating decomposition strategies like ours to ensure all entities are rendered.
>
> # [W2] Motivation: a fundamental data distribution problem
> > Questionable practical motivation with outdated baselines: Focuses on SD 1.5 (2022) while modern models (SD 3.5, Flux) with T5-XXL encoders likely handle long prompts well.
>
> > I cannot understand why we should use PromptDecomposer, if the use of SD 3.5 (or Flux -- I want this added as baseline comparsion too.) is already better in numbers.
>
> We thank the reviewer for raising this critical point regarding the motivation and practical value of our method. We agree that models like SD3.5 represent a significant leap forward; however, we respectfully argue that our approach is a generalizable framework and remains **highly relevant for SOTA models**, as it addresses **the fundamental data distribution challenge that persists regardless of the text encoder**.
> 1. **Text Encoder Capacity vs. Generative Prior:** While T5-XXL encoders allow longer sequences as inputs, the generative models are still predominantly trained on short, concise captions (e.g., LAION). Consequently, even if the encoder captures the full text, _the generative model lacks the learned distribution to render the complex narrative flow of a paragraph_. Our compositional approach addresses this specific data gap by leveraging the model’s existing capability to render concepts as components, regardless of the underlying architecture.
> 2. **Modern Models Are Improvable**: To demonstrate this, we successfully extended our method to **SD3.5 and Qwen-Image** in the new **Appendix C**.
> - **Empirical Gains:** As shown in **Table 5 & 6**, our method also improves modern baselines. For example, on **Qwen-Image** (uses Qwen2.5-VL as text-encoder), our method improves **Object Presence by +5.2%** and **Character Attributes by 3.69%**. PromptDecomposer-SD3.5 also outperforms the vanilla SD-3.5 (e.g., **+3.08% in Spatial Relation** and **+1.84% in Object Presence** on the DetailMaster benchmark).
> - **Qualitative Evidence: Figures 12 & 13** visualize how our prompt decomposition enables these models to capture more details (e.g., the presence of a "life guard" and seven different circles in Figure 12), demonstrating better adherence to complex, descriptive paragraphs from our compositional framework.
> 3. **Validity of SD-1.5 Baselines:** We primarily utilized SD-1.5 in the main paper to ensure **fair comparisons** with other methods (LLM4GEN, ELLA, LongAlign), which are all built on a SD-1.5 backbone. Comparing our method on SD3.5 against baselines on SD1.5 would have yielded unfair conclusions.

---

> ### Author Response · Authors · 2025-11-21
> **Official Response to Reviewer 2frZ (2)**
>
> # [W3] Transferability to modern architecture
> > Poor scalability to modern architectures: SD 3.5 experiments (Table 4) show minimal improvements despite requiring 1.2B parameters for PromptDecomposer-SD3.5
>
> We thank the reviewer for highlighting the parameter efficiency concern. We agree that the naive adaptation of our SD-1.5 architecture to SD-3.5 resulted in substantial parameter growth (1.2B) due to the increased cross-attention dimensionality.
>
> However, the limitation comes from the unmatched model size and computational resources. As discussed in the new **Appendix C.1** of our revision, we implement a **parameter-efficient PromptDecomposer** that amortizes trainable parameters into text encoders (e.g., T5-XXL in SD3.5) through LoRA,  and tune the text encoders to directly output decomposed representations. This optimization **lowers the parameters to only 160M** (a ~87% reduction), which makes PromptDecomposer-SD3.5 trainable on our device.
>
> As demonstrated in **Table 5** as well as the table below, this implementation can be trained efficiently and thus successfully generalizes to modern architectures. The PromptDecomposer-SD3.5 surpasses the base model on key metrics like _Object Presence_ (+1.84%) and _Spatial Relation_ (+3.08%). The improvements are even more pronounced on the stronger Qwen-Image baseline, where PromptDecomposer-Qwen **improves Object presence by 5.22% and character attributes by 3.69%**. While we observe a slight trade-off in _scene attributes_, probably due to the chunk operation on the text representations (please find more details in the **Appendix C**), the results validate that our compositional framework works across various architectures and is scalable to modern diffusion backbones.
>
> | Models | object presence | character attributes | character location | scene attributes | spatial |
> |:---|:---:|:---:|:---:|:---:|:---:|
> | SD-3.5-M | 31.19 | 30.37 | 26.69 | 91.64 | 28.90 |
> | PromptDecomposer-SD3.5 | **33.03** | **33.66** | **27.62** | 90.42 | **31.98** |
> | Qwen-Image | 31.63 | 33.89 | 31.01 | 93.35 | 37.14 |
> | PromptDecomposer-Qwen | **36.84** | **37.58** | **32.55** | 92.01 | **37.80** |
>
> # [W4] Ablation on text encoder
> > The approach appears limited to weak text encoders (CLIP), and generalization across different text-encoder architectures (CLIP vs T5 vs T5-XXL) needs systematic demonstration. Please add text-encoder comparison experiments for clear demonstration of your contribution.
>
> > Provide ablations showing improvements come from decomposition rather than just better text encoding (e.g., T5-XXL with SD 1.5 without decomposition)
>
> We thank the reviewer for this important question. We will address both concerns regarding encoder architectures and the specific source of performance improvements below:
>
> 1. **Generalization to Stronger Text Encoders (T5-XXL):** Our approach is not limited to CLIP-based models. As detailed in **Appendix C.1** and **Table 5 & 6**, we successfully adapted our method to **SD3.5** (which uses T5-XXL) and **Qwen-Image** (which uses Qwen2.5-VL).
>
> - **Results:** Even with these SOTA text encoders, our approach (PromptDecomposer-SD3.5 and PromptDecomposer-Qwen) consistently outperforms the base models on the DetailMaster benchmark (e.g., improving Character Presence on SD3.5 from 32.21% to 33.03% and Style from 86.47% to 92.14%).
>
> - This demonstrates that our **decomposition strategy provides gains orthogonal to the strength of the base text encoder**.
>
> 2. **Ablation: Decomposition vs. Better Encoding.** To prove that improvements stem from _composition_ rather than simply upgrading the encoder, we compared our method against a strict baseline that uses the **exact same text encoder (T5-XL) and same amount of parameters** but without decomposition.
>
> - **Setup:** As described in Section 4.4, the "No Composition" baseline in Figure 7 utilizes the same T5-XL encoder and identical parameter counts as our method, but processes the prompt in a single unitary query.
>
> - **Outcome:** As shown in the table below (derived from **Figure 7** data), the Compositional models (N=4, 5) significantly outperform the "No Composition" baselines, despite both using the same T5-XL encoder. This confirms that **the performance gain is driven by the compositional mechanism**, not the encoder upgrade.
>
> | Models | object presence | object location | character attributes | scene attributes |
> | :--- | :--- | :--- | :--- | :--- |
> | 4-components_noComp | 0.21 | 0.15 | 0.74 | 0.79 |
> | 4-components | 0.25 | 0.18 | *0.79 | 0.80 |
> | 5-components_noComp | 0.21 | 0.17 | 0.70 | 0.77 |
> | 5-components | 0.25 | 0.24 | 0.79 | 0.82 |

---

> ### Author Response · Authors · 2025-11-21
> **Official Response to Reviewer 2frZ (3)**
>
> # [W5] Comparing against SOTA models
> > …moreover, Table 4 reveals SD 3.5 already achieves strong scores (CLIPScore 34.97, DenScore 22.37) suggesting the problem may not exist for modern models. No experiments with Flux/SDXL provided.
>
> > Also, please add SD 3.5 / Flux vanilla (w/o tuning or adding PromptDecomposer component) to the main table for clear comparison.
>
> > Also for these SOTA models I think adding qualitative comparison might be helpful if the actual result of PromptDecomposer demonstrating the long-prompt is better than those SOTA models.
>
> > Add user studies comparing generation quality against modern models
>
> We thank the reviewer for this insightful comment. While we agree that modern models like SD3.5 achieve high scores on general aesthetic metrics, our results indicate they still struggle with the structural and semantic complexity of long prompts.
>
> 1. **Generalization Gap in SOTA Models:** As shown in **Table 5**, SD3.5 and FLUX-Dev still fail in fine-grained adherence. For instance, SD3.5 achieves only **26.69%** in Character Location. This demonstrates that while image quality (CLIPScore) is high, precise long-prompt adherence remains an open challenge.
>
> 2. **Enhancing SOTA Models:** To address the reviewer's concern regarding practical value against SOTA models, we have extended experiments on the SOTA **SD-3.5** and **Qwen-Image** in the revised paper to show the best results:
>
> | Models | object presence | character attributes | character location | scene attributes | spatial |
> |:---|:---:|:---:|:---:|:---:|:---:|
> | StableDiffusion-3.5 Medium | 31.19 | 30.37 | 26.69 | 91.64 | 28.90 |
> | Qwen-Image | 31.63 | 33.89 | 31.01 | 93.35 | 37.14 |
> | FLUX | 34.33 | 36.90 | 31.62 | 94.45 | 35.31 |
> | PromptDecomposer-SD3.5 | 33.03 | 33.66 | 27.62 | 90.42 | 31.98 |
> | PromptDecomposer-Qwen | **36.84** | **37.58** | **32.55** | 92.01 | **37.80** |
>
> - **Quantitative**: The table above report DetailMaster benchmark results, where **PromptDecomposer-Qwen achieves the best performance across all metrics except _scene attributes_. In the table below we evaluate the samples generated in this benchmark using image quality models, where the best results are obtained by the PromptDecomposer models except CLIPScore, which is known to be unreliable when assessing long-caption semantics. These results highlight the superior long-prompt following capability of PromptDecomposer-Qwen, justifying the practical value of our method in improving pre-trained T2I models.
>
> - **Qualitative:** We provide visual comparisons in **Figure 12 & 13**. These samples demonstrate that our method effectively captures multiple objects and their intricate attributes and spatial relationships that the base SOTA models miss (e.g., the presence of a "life guard" and seven different "circles" in Figure 12).
>
> | Models | CLIPScore | DenScore | PickScore | HPSv3 |
> | :--- | :--- | :--- | :--- | :--- |
> | StableDiffusion-3.5 Medium | 34.97 | 22.37 | 21.63 | 13.39 |
> | Qwen-Image | 33.85 | 22.25 | 20.98 | 8.556 |
> | FLUX-Dev. | 33.30 | 22.56 | 21.89 | 13.17 |
> | **PromptDecomposer-SD3.5** | 32.97 | **25.01** | 21.49 | **13.52** |
> | **PromptDecomposer-Qwen** | 34.12 | 22.93 | **22.04** | 12.05 |
>
> 3. **User Study**: We conducted a user study on 40 complex prompts (>400 tokens), where we shortlisted key words from the paragraph and prompted the user to choose the best sample in each batch based on semantic adherence and visual quality. As shown in **Figure 11, PromptDecomposer-Qwen** was preferred by users **23.4%** of the time, outperforming the strong FLUX-Dev baseline (18.3%) in terms of adherence to key concepts and human perception quality.

---

> ### Author Response · Authors · 2025-11-21
> **Official Response to Reviewer 2frZ (4)**
>
> # [W6] Evaluation metric
> > Evaluation Metrics : In Tab. 2, you reported HPSv3, but in Tab. 4, you reported HPSv2. Is it Typo? If it is typo, I see that PromptDecomposer or other baselines in Tab. 2 stays at around 6.7 (ELLA) to 13.26 (LongAlign), but using vanilla SD 3.5 shows 28.86. Pickscore and Denscore shows better numbers in PromptDecomposer or on some baselines, but difference is not as big compared to HPS difference.
>
> We thank the reviewer for their detailed examination of our manuscript. We apologize for the confusion caused by the inconsistency between Table 2 and Table 4. This was not a typo; we originally employed HPSv2 for the SD3.5 experiments.
> The reviewer is correct that the large score difference (from ~13 in Table 2 to ~28 in Table 4) is due to the different scaling of the metrics, rather than a disparity in model performance. To ensure a fair and direct comparison, we have re-evaluated our LoRA-adapted PromptDecomposer for SD3.5 using **HPSv3**. As shown in the updated table below, the HPSv3 score for PromptDecomposer-SD3.5 is **13.52**. This aligns with the range reported in Table 2 (e.g., LongAlign at 13.26 ), confirming that the previous gap was indeed a result of the metric change. We have updated Table 4 in the revised paper accordingly.
>
> | Models | CLIPScore | DenScore | PickScore | HPSv3 |
> | :--- | :--- | :--- | :--- | :--- |
> | StableDiffusion-3.5 Medium | **34.97** | 22.37 | **21.63** | 13.39 |
> | **PromptDecomposer-SD3.5** | 32.97 | **25.01** | 21.49 | **13.52** |

---

### Official Review · Reviewer_ZpWd · 2025-11-01

**Soundness:** 3
**Presentation:** 3
**Contribution:** 2
**Rating:** 4
**Confidence:** 5

**Summary:**

This paper addresses the challenge of generating high-quality images from lengthy text prompts by proposing PromptDecomposer, a trainable module that decomposes long prompts into manageable sub-prompts processed in parallel by pre-trained T2I models, with outputs fused via concept conjunction. The method achieves competitive performance on DetailMaster benchmark and demonstrates superior generalization on prompts exceeding 500 tokens, improving performance by 7.4% over existing approaches.

**Strengths:**

1. The paper proposes compositional long-text-to-image generation and unsupervised long-prompt decomposition methods to enable models to better perceive lengthy text inputs.

2. Experimental validation effectively demonstrates the effectiveness of the proposed approach.

**Weaknesses:**

1. While decoupling long text representations is a reasonable idea, the proposed modules bear strong resemblance to existing architectures (e.g., Q-Former), lacking sufficient novelty. The approach is heavily data-driven without explicit representation loss guidance, which raises concerns about the generalization capability of learned parameters. Additionally, the acquisition of image captions is critical for training but not thoroughly discussed.

2. In Table 1, SDXL-based models outperform the proposed method on Character Presence and Object metrics. Why were experiments not conducted on SDXL? SD-1.5 is outdated. As acknowledged in the limitations, transferring to SD3.5 does not yield significant improvements. Given the emergence of Flux, Qwen-Image, and similar models, exploring complex prompt generation on these newer architectures would be more valuable.

**Questions:**

How should this method be adapted to state-of-the-art models like Qwen-Image (with Qwen2.5-VL as encoder) or MetaQuery-type architectures? What modifications are necessary for effective transfer?

---

> ### Author Response · Authors · 2025-11-21
> **Official Response to Reviewer ZpWd (1)**
>
> We sincerely thank the reviewer for the positive and encouraging feedback. We appreciate the recognition of the compositional framework in processing the lengthy inputs, as well as the acknowledgement of the empirical results of our method in validating its effectiveness. We address the reviewer’s concerns in detail below.
>
> # [W1] Novelty
>
> > While decoupling long text representations is a reasonable idea, the proposed modules bear strong resemblance to existing architectures (e.g., Q-Former), lacking sufficient novelty.
>
> We appreciate the reviewer's feedback regarding architectural novelty. We acknowledge the resemblance to Q-Former, as we adopted this design for its proven **efficiency** and its **intuitiveness** in decomposing input representations into sequences of tokens. However, we wish to clarify that **our core novelty lies in the framework for long-prompt generation**, specifically:
>
> 1. **Unsupervised Prompt Decomposition:** Learning to break down long narratives into sub-prompt components optimized for the pre-trained T2I model's generation (guided solely by the frozen T2I model's gradients).
>
> 2. **Concept Conjunction:** These sub-prompts are then re-combined during the generation process to compose the output following the original long-prompt's narratives.
>
> To demonstrate that our framework is not tied to the Q-Former architecture, we conducted additional experiments across various diffusion backbones with different decomposition module design in **Appendix C**. Crucially, this compositional framework works effectively even with a simple module design that **add LoRA on Qwen2.5-VL**, significantly improved the Qwen-Image baseline (e.g., **HPSv3: 8.556 -> 12.05**), as detailed in **Appendix C** and the new **Table 5 & 6** . This confirms our approach functions as **a general framework where both the specific decomposition module and the T2I backbone are interchangeable**.
>
> | Models | CLIPScore | DenScore | PickScore | HPSv3 |
> | :--- | :--- | :--- | :--- | :--- |
> | Qwen-Image | 33.85 | 22.25 | 20.98 | 8.556 |
> | **PromptDecomposer-Qwen** | **34.12** | **22.93** | **22.04** | **12.05** |
>
> # [W2] Generalization
>
> > The approach is heavily data-driven without explicit representation loss guidance, which raises concerns about the generalization capability of learned parameters.
>
> We appreciate the reviewer's insightful comment regarding the unsupervised nature of our learning objective. We address the concerns regarding representation guidance and generalization below:
>
> 1. **Empirical Evidence Against Explicit Regularization:** We share the reviewer's interest in explicit representation guidance. In our preliminary experiments, we introduced a **similarity regularization loss** to force orthogonality among decomposed sub-prompts **for better semantic disentanglement**. However, as shown in the table below, imposing this explicit constraint **degraded performance** across all metrics compared to our unsupervised objective in Equation 5. This suggests that training the model with only the gradient from a frozen T2I backbone allows it to learn the generalizable decompositions, which are more effective than those shaped by rigid hand-crafted constraints.
>
> | Models | CLIPScore | DenScore | PickScore | HPSv3 |
> | :--- | :--- | :--- | :--- | :--- |
> | PromptDecomposer | 0.3519 | 0.2596 | 0.2308 | 12.61 |
> | w/ sub-prompts similarity regularization | 0.3457 | 0.2270 | 0.2133 | 10.01 |
>
> **2. Method Generalization:** Our approach yields superior generalization compared to baseline methods, particularly on inputs that contain significantly more tokens than that in the training data distribution:
>
>  - **Length Generalization:** As highlighted in **Figure 5 (Page 8)**, our model maintains robust performance on prompts exceeding 500 tokens—lengths significantly longer than the training data average. This confirms that the learned parameters capture transferable compositional structures rather than overfitting to specific training captions.
>
>  - **Architectural Generalization:** To further demonstrate the robustness of our method, we have successfully extended our experiments to DiT models SD3.5 and Qwen-Image, as detailed in the **Appendix C**. Notably on Qwen-Image, our PromptDecomposer can be trained efficiently (approx. 2500 steps) by applying LoRA on its text encoders, demonstrating that **the learning strategy generalizes** across different model architectures or diffusion paradigms.

---

> ### Author Response · Authors · 2025-11-21
> **Official Response to Reviewer ZpWd (2)**
>
> # [W3] Training data caption
>
> > Additionally, the acquisition of image captions is critical for training but not thoroughly discussed.
>
> We thank the reviewer for emphasizing the importance of training data transparency. While we briefly specified the use of the LongAlign dataset in Section 4.1, we clarify the specific composition and caption acquisition here. The dataset comprises approximately 2 million images sourced from SAM (500K), COCO2017 (100K), LLaVA (500K), and JourneyDB (1M). Crucially, to support long-prompt generation, these images were re-captioned using LLaVA-Next or ShareCaptioner to ensure dense, descriptive textual input. We have included these dataset details in **Lines 290-291 (Page 6)**.
>
> # [W4] Choice of base model
>
> > In Table 1, SDXL-based models outperform the proposed method on Character Presence and Object metrics. Why were experiments not conducted on SDXL? SD-1.5 is outdated.
>
> We thank the reviewer for this suggestion. We focused on SD-1.5 in Table 1 specifically to ensure a **fair, controlled comparison** with the primary baselines (ELLA, LongAlign, ParaDiffusion), all of which are built upon the SD-1.5 architecture. This allows us to isolate the performance gains attributable strictly to our compositional approach rather than a stronger backbone.
>
> However, we fully agree that demonstrating scalability on modern architectures is critical. To address this, we extended our evaluation **beyond SDXL** to the modern DiT models, which represent the current state-of-the-art. New experiments in **Appendix C (Table 5)** demonstrate our method’s efficacy on **SD3.5 and Qwen-Image**. For instance, our **PromptDecomposer-SD3.5** outperforms the baseline SD3.5 model on complex _spatial relations_ and _character attributes_. These results confirm that our compositional strategy also generalizes effectively to next-generation backbones.
>
> # [W5 & Q1] Transferability to modern architectures
>
> > As acknowledged in the limitations, transferring to SD3.5 does not yield significant improvements. Given the emergence of Flux, Qwen-Image, and similar models, exploring complex prompt generation on these newer architectures would be more valuable.
>
> > How should this method be adapted to state-of-the-art models like Qwen-Image (with Qwen2.5-VL as encoder) or MetaQuery-type architectures? What modifications are necessary for effective transfer?
>
> We thank the reviewer for the constructive suggestion to explore state-of-the-art architectures. We agree that transferability is vital. To address the parameter scaling limitations observed in our initial SD3.5 experiments (where the module grew to 1.2B parameters), we have implemented a **parameter-amortized solution** that yields significant improvements on SD3.5 and Qwen-Image.
>
> 1. **Module Adaptation:** As detailed in **Appendix C.1**, we leverage the reasoning capabilities of the strong text encoders (e.g., Qwen2.5-VL, T5-XXL) in modern T2I architectures to generate decomposed representations directly:
>
> - **Decoder-Only Architectures (e.g., Qwen-Image):** We replicate the input tokens N times and prepend learnable component tokens {<|comp_0|>,...,<|comp_N-1|>} to each segment (as illustrated in **Figure 10**). The model processes this expanded prompt as a single contiguous sequence, and we extract the output states corresponding to the component tokens as our decomposed representations.
>
> - **Encoder-Based Architectures (e.g., SD3.5, FLUX):** We apply LoRA to the text encoder (e.g., T5-XXL) and split the output into decomposed embeddings directly. For SD3.5 specifically, PromptDecomposer processes the CLIP representation separately and concatenates it with the T5 outputs to match its multi-encoder design.
>
> 2. **Empirical Results:** This LoRA-based adaptation reduces trainable parameters from **1.2B to 160M** (a ~10x reduction compared to our initial SD3.5 attempt). As shown in **Table 6**, applying our method to Qwen-Image **improves HPSv3 from 8.556 to 12.05**, demonstrating that our compositional approach scales successfully to modern architectures.
>
> | Models | CLIPScore | DenScore | PickScore | HPSv3 |
> | :--- | :--- | :--- | :--- | :--- |
> | StableDiffusion-3.5 Medium | 34.97 | 22.37 | 21.63 | 13.39 |
> | **PromptDecomposer-SD3.5** | 32.97 | **25.01** | 21.49 | **13.52** |
> | Qwen-Image | 33.85 | 22.25 | 20.98 | 8.556 |
> | **PromptDecomposer-Qwen** | **34.12** | **22.93** | **22.04** | **12.05** |

---

> ### Comment · Reviewer_ZpWd · 2025-11-28
>
> I appreciate the authors’ responses, which address most of my concerns. I will raise my score accordingly.

---

> > ### Author Response · Authors · 2025-11-28
> > **Official Response to Reviewer ZpWd**
> >
> > We sincerely thank the reviewer for their engagement during the discussion phase and for reconsidering their evaluation. We are glad to hear that our response and additional experiments have successfully addressed your concerns in:
> >
> > 1. **Novelty**. Our novelty lies in a compositional framework that allows pre-trained T2I models to generalize to longer prompts.
> > 2. **Generalizability**. Our method is a general framework applicable across various diffusion backbones including UNet/DiT and diffusion/flow-matching. Moreover, our method is not constraint to the Q-Former architecture, which we chose as the most intuitive module design to our purpose.
> > 3. **Transferability**. Our compositional generation method further improves SOTA models' performance on extreme long prompts that rarely seen in their training data. We demonstrate empirical gains both quantitatively and qualitatively through additional experiments on SD-3.5 and Qwen-Image.
> >
> > Thank you again for your encouraging feedback! Those constructive comments helped improve the clarity and quality of our work. We remain available for the remainder of the discussion period should you have any further questions or require additional clarifications. We will ensure that all the discussed revisions are incorporated into the final version of the paper.

---

### Official Review · Reviewer_TWGK · 2025-11-01

**Soundness:** 3
**Presentation:** 3
**Contribution:** 3
**Rating:** 6
**Confidence:** 4

**Summary:**

The paper aims to enhance image generation on long, paragraph-length prompts. It proposes a compositional pipeline with a trainable PromptDecomposer that splits a long prompt into distinct sub-prompts. A pre-trained T2I model processes these sub-prompts in parallel, and outputs are merged via concept conjunction. The method shows good gain on >500-token prompts in the DetailMaster benchmark.

**Strengths:**

- Good motivation and problem formulation: it’s a well-established research question that the state-of-the-art image generation models are usually trained on limited-length captions, which generates lower-quality images on long prompts.
 - Intuitive modules and good results: the proposed modules in PromptDecomposer are well motivated. The experiments show PromptDecomposer clearly outperformed baselines on long prompts.

**Weaknesses:**

- Limited novelty: many modules in this paper can be found in references. For example, cross-attention, T5, CLIP are off-the-shelf modules. It’ll be great if the authors could explain more about what’s the unique contribution and novelty in this paper.
 - Risk of losing global coherence: when merging independently generated components,  global coherence (such as lighting, perspective, style) might be lost, or might be conflicting with other components (e.g. day vs night, mountain vs sea).

**Questions:**

- The authors ablated the number of learnable queries in the paper. But they are still fixed. I wonder if we should make the number of learnable queries adaptive or dependent on the prompts?

---

> ### Author Response · Authors · 2025-11-21
> **Official Response to Reviewer TWGK (1)**
>
> We sincerely thank the reviewer for the positive and encouraging feedback. We appreciate the recognition of the fundamental long-prompt generalization problem for T2I models, as well as the acknowledgement of the intuitiveness and effectiveness of our method. We address the reviewer’s concerns in detail below.
>
> # [W1] Novelty
>
> > Limited novelty: many modules in this paper can be found in references. For example, cross-attention, T5, CLIP are off-the-shelf modules. It’ll be great if the authors could explain more about what’s the unique contribution and novelty in this paper.
>
> We sincerely thank the reviewer for the opportunity to clarify our contributions. While we utilize established components (e.g., Cross-Attention, T5), our core novelty lies not in these individual modules, but in the **compositional framework**, which enables pre-trained T2I models to handle long, descriptive paragraphs via **unsupervised prompt decomposition**.
>
> To demonstrate that this is a generalized framework not constraint to the specific module design nor the diffusion backbones, we have added additional experiments in the revision (see **Appendix C**  and the new **Table 5 & 6**). We successfully extended our approach to **StableDiffusion-3.5** and **Qwen-Image**, covering both UNet/DiT architectures and Diffusion/Flow-matching paradigms.
>
> | Models | CLIPScore | DenScore | PickScore | HPSv3 |
> | :--- | :--- | :--- | :--- | :--- |
> | StableDiffusion-3.5 Medium | 34.97 | 22.37 | 21.63 | 13.39 |
> | **PromptDecomposer-SD3.5** | 32.97 | **25.01** | 21.49 | **13.52** |
> | Qwen-Image | 33.85 | 22.25 | 20.98 | 8.556 |
> | **PromptDecomposer-Qwen** | **34.12** | **22.93** | **22.04** | **12.05** |
>
> As shown, our compositional framework works effectively **regardless of the architecture design in the decomposition module**. For instance, implementing our compositional objective via **a simple LoRA on the Qwen2.5-VL** text encoder improves Qwen-Image’s long-prompt generation performance on **HPSv3 from 8.556 to 12.06**. Overall, we intentionally selected simple module designs (like the Q-Former and LoRA) to highlight that the performance gains stem from the compositional strategy itself, rather than complex network engineering. We also provide a quantitative comparisons for these results in **Figure 12 & 13**, as well as a user study on the long-prompt adherence in **Figure 11**.
>
> # [W2] Global semantic coherence
>
> > Risk of losing global coherence: when merging independently generated components, global coherence (such as lighting, perspective, style) might be lost, or might be conflicting with other components (e.g. day vs night, mountain vs sea).
>
> We thank the reviewer for this insightful comment. We acknowledge that compositional generation often faces the risk of disjointed global attributes (e.g., conflicting lighting or perspective). However, our method is designed specifically to overcome this through **end-to-end global optimization**, rather than hard linguistic splitting.
>
> 1. Unlike methods that split sentences (which loses global context ), our PromptDecomposer learns to decompose representations by minimizing the difference between the summed sub-prompt noise predictions and the global noise prediction (Eq. 5 ). Because the loss is calculated against the global prompt's output, the decomposed sub-prompts are mathematically constrained to share global information (such as style and lighting) to minimize reconstruction error. If sub-prompts implied conflicting perspectives, the reconstruction loss would increase.
>
> 2. **Evidence: Shared Global Context.** This shared global context is visually evident in **Figure 8**. The individual generation from each sub-prompt still renders the same background (light blue rug, black floor) and lighting conditions without merging.
>
> 3. **Ablation: The Necessity of Shared Information.** To validate this, we conducted an additional experiment where we applied **semantic regularization** to force the sub-prompts to be semantically distinct (orthogonal), thereby reducing their shared global context. As shown in the table below, penalizing shared information degraded image quality (HPSv3 dropped from 12.61 to 10.01). This confirms that our model naturally learns to distribute global coherence (lighting, style) across all sub-prompts to ensure a seamless merge.
>
> | Models | CLIPScore | DenScore | PickScore | HPSv3 |
> | :--- | :--- | :--- | :--- | :--- |
> | PromptDecomposer | 0.3519 | 0.2596 | 0.2308 | 12.61 |
> | w/ sub-prompts similarity regularization | 0.3457 | 0.2270 | 0.2133 | 10.01 |

---

> ### Author Response · Authors · 2025-11-21
> **Official Response to Reviewer TWGK (2)**
>
> # [Q1] Adaptive decomposition
> > The authors ablated the number of learnable queries in the paper. But they are still fixed. I wonder if we should make the number of learnable queries adaptive or dependent on the prompts?
>
> We thank the reviewer for this insightful suggestion. We agree that an adaptive mechanism determining the number of components (N) based on prompt complexity is a promising direction for optimizing computational efficiency.
> However, regarding generation quality, our empirical results demonstrate that a fixed decomposition strategy is highly robust, even for shorter prompts where N might exceed the semantic granularity. To verify this, we evaluated our method on **T2I-CompBench and GenEval**, which primarily feature concise prompts (eg., "a photo of a dog." ). As shown in the table below, our method (PromptDecomposer) incurs **no performance degradation** on these benchmarks. In fact, it achieves best performance in attribute binding (e.g., **0.7113** in Color and **0.5204** in Shape), significantly outperforming the SD-1.5 and other baselines.
> | Models | color | shape | texture | spatial | numeracy | GenEval |
> |:---|:---:|:---:|:---:|:---:|:---:|:---:|
> | SD-1.5 | 0.3647 | 0.3768 | 0.4095 | 0.5064 | 0.3197 | 0.4418 |
> | LLM4GEN | 0.5084 | 0.4167 | 0.5085 | **0.6254** | **0.3828** | 0.4083 |
> | ELLA | 0.6269 | 0.4250 | 0.5585 | 0.5713 | 0.3013 | 0.4971 |
> | LongAlign | 0.5654 | 0.4693 | 0.5259 | 0.5698 | 0.3683 | **0.5075** |
> | PromptDecomposer | **0.7113** | **0.5204** | **0.6253** | 0.6015 | 0.3701 | 0.4960 |
>
> This suggests that our learned decomposition effectively handles varying prompt lengths without "over-decomposing" simple inputs. We have added these results on standard T2I benchmarks in **Appendix B and Table 4**, as well as the discussion regarding adaptive decomposition as a future direction in the revised **Section 5, Page 9 & 10**.

---

### Author Response · Authors · 2025-11-21
**General Response to All Reviewers**

We thank the reviewers for their constructive feedback and positive outlook. We are glad that the reviewers recognized the significance of the **problem formulation** (TWGK) and found our **energy-based compositional framework** to be interesting and effective (TWGK, ZpWd, 2frZ). In particular, we appreciate the acknowledgment of our **latent-space decomposition strategy** in avoiding semantic fragmentation (rbyh), as well as the recognition of our model's **training efficiency and comprehensive experimental validation** (ZpWd, 2frZ, rbyh).

We truly appreciate all reviewers and meta reviewer’s time and effort. **We have carefully read and addressed all your concerns, including:**
1. Transferability to modern architectures (2frZ, ZpWd, rbyh);
2. Insufficient evaluations (2frZ, ZpWd, rbyh);
3. Practical value with SOTA models (2frZ, rbyh);
3. Limited novelty in module designs (TWGK, ZpWd);
4. Risk of losing global coherence (TWGK);
5. Possibility and necessity of an adaptive decomposition (TWGK).

**All major modifications are highlighted in blue in the paper.** We thank you again for your time and constructive insights.

---

### Author Response · Authors · 2025-12-01
**Rebuttal summary for AC**

# Reviewer ZpWd (Scalability & Novelty)
1. **Scalability issue resolved via LoRA.** The reviewer noted that our original module design was too heavy for large models (1.2B params for SD3.5). We introduced a LoRA implementation of our method (Section 3.4 and Appendix A.1), **reducing params by ~90% (to 160M)** on large-scale models, and making it trainable on our device to show improvements.
2. We clarified **our novelty as the compositional framework** instead of specific module designs. This framework can be generalized across architectures (UNet/DiT) and paradigms (diffusion/flow-matching). To show, we ran **experiments on SD-3.5 and Qwen-Image**, which significantly improved generalization to long-prompts (**Character Presence +5.2%, Character Attributes +3.69%, HPSv3: 8.556 &rarr; 12.05** on Qwen-Image; **Spatial Relation +3.08%, Object Presence +1.84%** on SD-3.5).
3. Concerns about the comparison fairness are all resolved. In the original paper we noted the adopted train set, we've added more details on image captions (lines 310-311). In the rebuttal, we also explained the choice of the base model was for fair comparisons.

**Net effect:** The reviewer has explicitly acknowledged these improvements and **raised their score** accordingly, noting that their concerns regarding **transferability and novelty are addressed.**
# Reviewer TWGK (Novelty & Soundness)
1. **Novelty**: We provide **a similar explanation to reviewer ZpWd**, clarifying our novelty as the compositional framework. We implemented our framework utilizing those well established modules for their proven efficiency.
2. **Global coherence concerns addressed.** We provided **an ablation study** showing that explicit semantic separation hurts performance (Table in rebuttal), proving that our training objective naturally forces sub-prompts to share global context to minimize training loss.
3. We acknowledged the question about **adaptive decomposition as a promising future direction**, but also provided **evaluations on standard T2I benchmarks** to show our fixed decomposition is robust across different scenarios.

**Net effect:** The reviewer’s main weakness on Novelty is addressed, as **our rebuttal resolved the same concern of reviewer ZpWd to raise their score**. The other weakness stems from misunderstanding and is clarified through empirical evidence. Meanwhile, their positive assessments (problem formulation, intuitive method and solid results) remain intact.
# Reviewer 2frZ (Motivation & Scalability)
1. **Clarify Motivation:** The reviewer questioned if modern models even struggle with long-prompts. We **benchmarked SD-3.5, Qwen-Image and FLUX** and include them as baselines in Table 1. Results showed that even SOTA model **misses more than a half of characters** in dense paragraphs. This justifies long-prompt generalization as a fundamental challenge regardless of the advancement in architectures.
2. **Scalability:** We provide **a similar explanation to reviewer ZpWd**, and ran experiments to show a LoRA implementation of our method **improves large-scale Qwen-Image and SD-3.5 models**. This also demonstrates the **practical value** of our approach in enhancing SOTA models, **achieving best results on multiple metrics** on the DetailMaster benchmark.
3. The concern on the source of performance gains is also resolved. The experiments of the LoRA implementation improved Qwen-Image without changing its text-encoder. We also validated the performance gains come from decomposition by isolating the effect of text encoder **through ablation studies**.
4. The concern on the evaluation metrics resolved through updating the corresponding table.

**Net effect:** As requested by the reviewer to raise their score, we first showed SOTA models fail on long-prompts. Then showed how our method could be applied to enhance such performance, validated through both quantitative evaluations and qualitative user studies. Finally, we showed these gains come from decomposition instead of merely upgrading text-encoder through ablation studies. We believe all the concerns are resolved, as these improvements are either empirical evidence or acknowledged by other reviewers.
# Reviewer rbyh (Scalability & Evaluation)
1. **Scalability:** We provide **a similar explanation to reviewer ZpWd**.
2. We extend the evaluations to **include LongAlign eval-set and two standard T2I benchmarks**, showing our method's robustness on normal prompts and the consistent improvements across benchmarks. We also added a layout-control method as **a training-free baseline**.
3. We addressed the concerns on train-set and computational efficiency by adding details in **lines 310-311 and Appendix A.3**.

**Net effect:** Our revision resolved the reviewer’s main concern on Scalability, which **resolved the same concern of reviewer ZpWd**. We extended evaluations and added more details to address other weaknesses. We believe these improvements now clearly favors an accept recommendation.

---

### Author Response · Authors · 2025-12-02
**Summary after the Discussion Period**

We thank the reviewers and ACs for their careful assessment and have substantially revised the paper to address all substantive concerns. In summary, the **Scalability of our method** is the major concern shared across reviewers ZpWd, 2frZ, rbyh. We presented a LoRA implementation of our method, and **ran experiments on SD-3.5 and Qwen-Image** to show how our method could be applied to improve large-scale SOTA models. These additional results are updated in **Table 1, 2 and Figure 7**. Besides, we have provided empirical evidence or detailed explanations to address other weaknesses and questions, including novelty, our motivation, method soundness, dataset details and expanding evaluation scope.

In doing so, we show that several stated weaknesses stemmed from misunderstandings or from points already present in the paper, which we now make explicit via new experiments (notably the significant improvements on Qwen-Image), additional analyses, and clearer exposition. Taken together, the updated evidence strongly supports our method as a general framework to enhance pre-trained T2I models' performance on prompts longer than what they see in training data.

---

### Meta-Review · Area_Chair_WWkd · 2026-01-06

**Summary:**

This paper proposes PromptDecomposer, a compositional approach for enabling pre-trained text-to-image (T2I) diffusion models to handle long, paragraph-length text prompts. The method decomposes long prompts into manageable sub-prompts in the latent space, processes them in parallel through a frozen pre-trained T2I model, and merges the outputs via concept conjunction.

The initial reviews raised five major concerns:
- Scalability to modern architectures. The method appeared limited to SD-1.5 (outdated), raising questions about practical applicability to SOTA models like FLUX, SD3.5, and Qwen-Image.
- Practical motivation with modern models. Do SOTA models with advanced text encoders (T5-XXL, Qwen2.5-VL) actually struggle with long prompts?
- Source of performance gains. Are improvements due to the compositional framework or simply from using better text encoders (e.g., T5-XL vs CLIP)?
- Limited novelty. The modules (Q-Former, cross-attention, T5, CLIP) are off-the-shelf components.
- Inference cost: What is the inference cost of the method compared with its baseline counterpart?

**Reviewer Concerns:**

Reviewer concerns that were addressed:
- Scalability to Modern Architectures. Authors implemented a parameter-efficient LoRA-based variant and demonstrated substantial improvements on Qwen-Image (HPSv3: 8.556→12.05; Character Presence: +5.2%) and SD-3.5 (Spatial Relation: +3.08%).
- Practical Motivation with Modern Models. The authors provided compelling evidence that even SOTA models fail significantly: FLUX-Dev achieves only 34.33% Character Presence (missing 65% of described objects), and SD-3.5 achieves 26.69% Character Location accuracy.
- Source of Performance Gains. The authors provided ablation studies isolating decomposition effects: "No Composition" baseline using identical T5-XL encoder and parameters significantly underperforms (74% vs 79% character attributes).

Reviewer concerns that are still outstanding:
- Limited Novelty. The authors clarified that novelty lies in the compositional framework for unsupervised prompt decomposition, not individual modules. But the proposed compositional framework also closely aligned with previous compositional generation methods, such as https://arxiv.org/abs/2004.06030 and https://arxiv.org/abs/2206.01714.
- Inference cost: Compositional generation increases FLOPs by ~4x theoretically, translating to ~2x wall-clock time due to parallelization (10 iter/s vs 22 iter/s for LongAlign).

**Reviewer Scores:**

Reviewer TWGK: 6,
Reviewer 2frZ: 6,
Reviewer ZpWd: 6,
Reviewer rbyh: 6

---

### Decision · Program_Chairs · 2026-01-26

Accept (Poster)